# DKRF: Dynamic Knowledge Reasoning for Out-of-Distribution Generalization in Mobile GUI Agents

## Abstract

Graphical User Interface (GUI) agents demonstrate significant potential in automating GUI tasks, yet their performance often drops sharply when facing out-of-distribution (OOD) scenarios (e.g., unseen task, different layout, etc.) in the open world. Previous methods, modular agent frameworks and end-to-end native agents, are designed based on in-distribution (ID) mobile data, whether through manual designed modules or specially collected training sets, while neglecting the adaptability to diverse data in potential OOD mobile scenarios. To overcome these limitations, we propose Dynamic Knowledge Reasoning Fine-tune (**DKRF**), a paradigm that shifts the agent's core capability from memorizing ID patterns to reasoning dynamically with external knowledge. During training, the model *explicitly* receives dynamic knowledge (e.g., *trajectories of similar tasks* or *reusable meta-functions*) and need to *incorporate* this knowledge in its reasoning chain, thereby learning to make knowledge-driven decisions. Based on DKRF, 1) we train an end-to-end native agent, **DKR-GUI**, and 2) further propose a modular agent framework, **MA-DKR**, which uses DKR-GUI as the planning core combined with knowledge retrieval and an executing agent to achieve collaboration between complex reasoning and precise execution. Experiments on multiple mobile benchmarks show that both DKR-GUI and MA-DKR significantly outperform existing methods, achieving an average 9.2% improvement in success rate in OOD mobile scenarios while also maintaining state-of-the-art performance in ID mobile tasks. Our results demonstrate that dynamic knowledge reasoning provides a general and effective solution for OOD generalization, highlighting its potential as a foundation for robust, knowledge-driven interactive agents.

## 1 Introduction

Graphical User Interface (GUI) agents, particularly those powered by Multimodal Large Language Models (MLLMs) (Team et al., 2024; Hurst et al., 2024; Wang et al., 2024c; Bai et al., 2025; Liu et al., 2023; 2024), can perform complex, cross-application tasks on mobile devices solely by analyzing screenshots. As a key component of next-generation human-machine interaction, they promise to simplify digital operations and boost productivity, and are typically implemented as either modular agent frameworks or end-to-end native agents. However, the development of these vision-based agents relies on pre-collected data, which is used to either manually design modules or fine-tune the MLLMs. Since such data cannot encompass the vast complexity of real-world mobile GUI environments, these methods inherently lack the adaptability required for out-of-distribution (OOD) mobile scenarios (e.g., unseen app, novel task, etc.) (Sun et al., 2025).

A classic implementation method is the modular agent framework, which decomposes tasks into high-level planning and low-level perception components, and uses the general capabilities of (M)LLMs for planning task, while integrating external tools (Wang et al., 2024b; 2025a; Zheng et al., 2024; Gou et al., 2024; Zhang et al., 2025a). Although interpretable and flexible, their reliance on pre-defined workflows makes them unreliable when encountering unexpected mobile situations. The other route involves end-to-end native agents, which learn to directly map inputs to actions by training on large-scale trajectory data (Wu et al., 2024b; Qin et al., 2025; Xu et al., 2024; Zhang et al., 2025b). However, this "closed-set" learning paradigm leads to issues of fixed knowledge, making

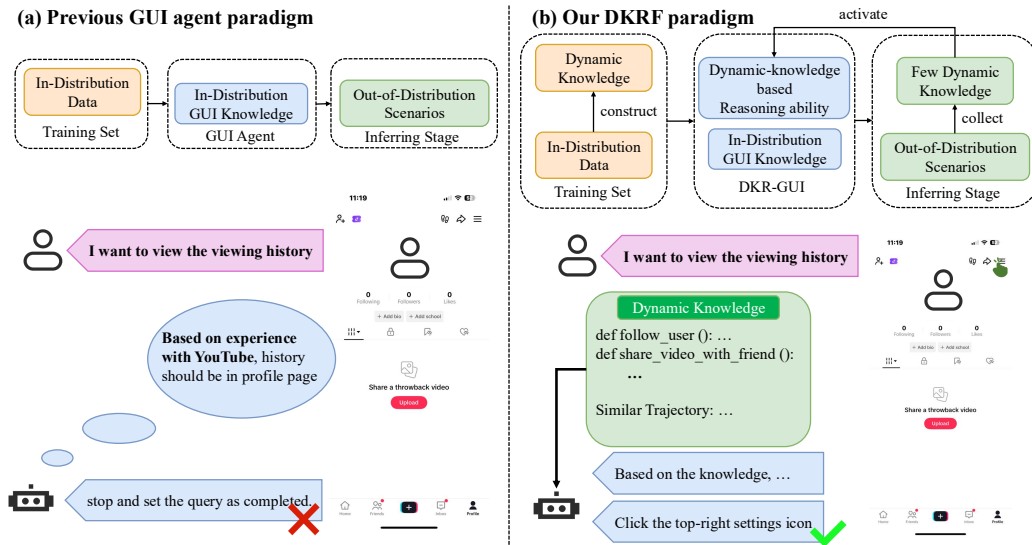

Figure 1: **A comparison of different methods.** Previous methods, constrained by a closed training set, struggle to effectively leverage the few knowledge available in OOD mobile scenarios. In contrast, DKRF acquires the ability to reason with dynamic knowledge during training, which can then activate in OOD mobile scenarios to effectively utilize this knowledge without updating or manual design.

model updates costly and risky. Both modular agent frameworks and end-to-end native agent models struggle with adaptability in the face of diverse OOD mobile scenarios. The core problem is that in a dynamic mobile environment, it is impractical to either collect massive amounts of data to cover all scenarios or to frequently update the model after encountering new ones. Consequently, a more rational technical paradigm involves the effective utilization of mobile GUI knowledge excluded from the training distribution. Such knowledge encompasses sources including user interaction history, application documentation, etc. Furthermore, this approach offers a more privacy-preserving alternative.

We argue that the key to solving the generalization problem lies in fundamentally reshaping the learning objective: shifting from merely memorizing action patterns to learning a generalizable capability for reasoning with external knowledge. To this end, we propose Dynamic Knowledge Reasoning Fine-tune (DKRF) in mobile GUI agents. Specifically, we utilize in-distribution (ID) mobile datasets of base model, such as Android Control (Li et al., 2024a) and AITZ (Zhang et al., 2024), as our training set and sample a collection of trajectories from them. For each sample, in addition to using the instructions and screenshots as inputs, we retrieve a few semantically similar trajectories from the training set to serve as dynamic knowledge. Furthermore, we employ a stronger MLLM to generate a reasoning on this dynamic knowledge, which acts as the ground-truth for the output. By mixing this knowledge-augmented data with the original data, we train an end-to-end native agent (DKR-GUI) that learns dynamic reasoning. The capability to reason over external knowledge, which DKR-GUI acquires during training, can then be activated by corresponding dynamic knowledge in OOD mobile scenarios.

To synthesize the respective advantages of end-to-end native agents and modular frameworks, we also design the MA-DKR framework and further enhance its overall performance in complex mobile tasks. This is a modular architecture which use DKR-GUI as the planning core, responsible for high-level complex reasoning and strategy formulation. DKR-GUI decomposes high-level instructions into a series of simple, executable low-level instructions, which are then passed to an executing agent specializing in precise visual grounding and interaction. This design, which separates complex reasoning from precise execution, allows different modules to perform their specialized functions, leading to a significant improvement in the overall system's performance and robustness.

Our main contributions are:

1. We propose DKRF paradigm, which shifts the mobile agent's core capability from pattern memorization to dynamic knowledge based reasoning. This ability is acquired during the training phase and can activate in OOD mobile scenarios.

2. Based on this paradigm, we train a powerful end-to-end native agent, DKR-GUI. Furthermore, by integrating DKR-GUI with knowledge management and executing agents, we introduce the MA-DKR framework, which enhances performance in both ID and OOD mobile scenarios.

3. Extensive experiments validate the effectiveness of our methods. MA-DKR achieves state-of-the-art (SOTA) performance on multiple mobile GUI benchmarks (e.g., 72.9% SR on Android Control). The inclusion of DKR-GUI as planning agent improves the performance of executing agents by an average of 4.1% on ID tasks and 9.2% on OOD mobile tasks.

## 2 RELATED WORK

### 2.1 GUI AGENTS

Research on GUI agents has advanced significantly, mainly following two approaches: modular agent frameworks and end-to-end native agents. While both show promise, they have limitations when facing dynamic OOD mobile scenarios.

Modular agent frameworks are driven by the powerful capabilities of commercial (M)LLMs like GPT-4 (Achiam et al., 2023), Gemini (Team et al., 2024). They work by decomposing a complex GUI task and using the (M)LLMs for high-level planning, reasoning, and perception, while relying on other modules or tools for final actions (Zheng et al., 2024; Wu et al., 2024a; Yan et al., 2023; Sumers et al., 2023). To improve performance, researchers have focused on "framework engineering," such as adding memory modules (Wang et al., 2025a; Jiang et al., 2025; Wang et al., 2025b), specialized visual components (Gou et al., 2024; Wu et al., 2025; Cheng et al., 2024; Tang et al., 2025) or using multi-agent collaboration (Wang et al., 2024b;a; 2025a).

End-to-end native agents aim to unify perception, reasoning, and action into a single model. This is typically done through supervised fine-tuning (SFT) or reinforcement learning (RL) on large datasets of "screenshot-instruction-action" samples (Wu et al., 2024b; Qin et al., 2025; Xu et al., 2024; Luo et al., 2025; Liu et al., 2025b; Lin et al., 2025; Chen et al., 2025; Zhang et al., 2025b; Sun et al., 2024; Hong et al., 2024). The model learns to predict the next action directly from the visual inputs.

While both approaches have achieved some success, their design processes do not adequately consider the agent's generalizability in OOD mobile scenarios. Modular agent frameworks rely on manual-designed modules. Although flexible and extensible, their pre-defined rules often fail in OOD settings, typically requiring human intervention to broaden their applicability. End-to-end native agents, which are based on large, pre-collected datasets, reduce manual effort but lack the adaptability to few samples. Furthermore, attempting to frequently update them for OOD scenarios carries the risk of "catastrophic forgetting."

To address these issues, we propose DKRF, which shifts the mobile agent's objective from memorizing patterns in the training set to reasoning over external knowledge. We find that the ability to reason with dynamic knowledge, acquired ID, can also be activated in OOD scenarios. This enables DKR-GUI to adapt more effectively to OOD mobile environments.

### 2.2 THE USE OF DYNAMIC KNOWLEDGE IN GUI

To address the adaptability problem, some recent studies have started to use dynamic knowledge. This includes building memory systems to retrieve past successful experiences (Liu et al., 2025a; Jiang et al., 2025; Sun et al., 2025), creating reusable high-level "skills" from common operation sequences (Wang et al., 2025a;b; Zheng et al., 2025; Wang et al., 2024d) or directly searching for information on the internet (Agashe et al., 2024).

However, these methods are mainly modular agent frameworks that rely on closed-source MLLMs to process and leverage dynamic knowledge. Their research focus is primarily on how to acquire and retrieve knowledge, rather than on cultivating the agent's intrinsic capability to reason with and act upon this information.

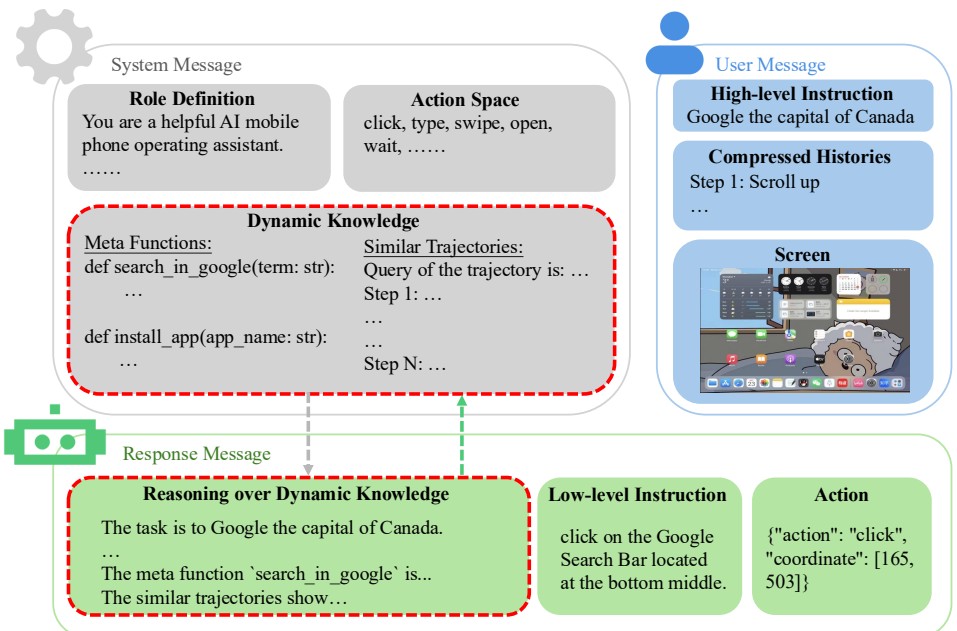

Figure 2: **An overview of our Dynamic Knowledge Reasoning Fine-tune (DKRF) paradigm.** This paradigm redefines the agent's learning objective. Instead of mapping observations directly to actions, the model processes not only the screen and instruction, but also dynamically external knowledge (e.g., Meta Functions, Similar Trajectories). It then generates an explicit reasoning process that utilizes this knowledge, followed by a low-level instruction and the final action. This forces the model to learn how to reason with knowledge rather than just memorizing action patterns.

In contrast, we propose DKRF to directly improve the MLLM's ability to utilize such knowledge, enhancing the GUI Agent's performance in both ID and OOD scenarios.

## 3 METHOD

The core of our work is the DKRF paradigm. Based on this paradigm, we develop a new end-to-end native agent, DKR-GUI, which can function as a powerful independent agent. To further enhance its capabilities, we also introduce the MA-DKR framework, which uses DKR-GUI as its central planning agent, supported by a knowledge retriever and an executing agent.

### 3.1 PROBLEM FORMULATION

We formulate the task of a GUI agent as a sequential decision-making process. Since the agent can only observe a visual screenshot of the GUI at each timestep without knowing the complete underlying state, the process can be modeled as a partially observable markov decision process (POMDP), defined as $\mathcal{M} = (\mathcal{S}, \mathcal{O}, \mathcal{A}, \mathcal{T})$, where $\mathcal{S}$ is the state space (current state of the mobile device), $\mathcal{O}$ is the observation space (instructions $\mathcal{I}$, screenshots, etc.), $\mathcal{A}$ is the action space (e.g., click, type, swipe), $\mathcal{T} : \mathcal{S} \times \mathcal{A} \to \mathcal{S}$ is the state transition function.

At each timestep $t$, the agent receives an observation $o_t \in \mathcal{O}$ and generates a sequence of steps, each containing a thought $t_t$ and an action $a_t$. The agent's task is to learn a policy $\pi$ that auto-regressively predicts the joint probability distribution of the current thought and action, given the history:

$$P(t_n, a_n | \mathcal{I}, o_1, t_1, a_1, ..., o_{n-1}, t_{n-1}, a_{n-1}, o_n). \tag{1}$$

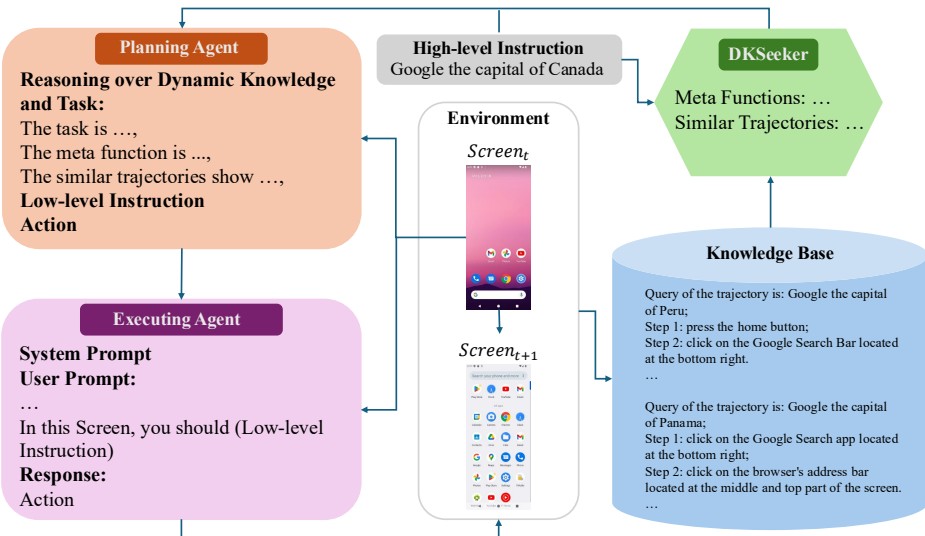

Figure 3: **The architecture of the MA-DKR framework, which illustrates a clear division of labor.** The DKSeeker retrieves relevant knowledge from a knowledge base. This knowledge passes to the planning agent (DKR-GUI), which specializes in complex reasoning and generating a simple, low-level instruction. Finally, the executing agent, a SOTA model with strong grounding capabilities, focuses solely on carrying out this low-level instruction. This modular design maximizes both planning robustness and execution precision.

## 3.2 THE DYNAMIC KNOWLEDGE REASONING FINE-TUNE (DKRF) PARADIGM

Our method center around the DKRF paradigm. Instead of training a model to directly map observations to actions, DKRF redefines the learning objective. For each training sample, the paradigm provides the model with relevant external knowledge $D_k = \{d_1, d_2, ..., d_k, f_m\}$, where $D_k$ denotes the dynamic knowledge, $d_k$ denotes a similar trajectory within the dynamic knowledge, and $f_m$ denotes the meta-function summarized from these similar trajectories. The model is then trained to map the combination of observations and this dynamic knowledge to thoughts and actions: $(t_n, a_n) \sim \pi_{DKR}(o_n | \mathcal{I}, D_k, history)$, as shown in figure 2.

A key aspect of DKRF is how the training data is structured. The ground-truth thought, $t^*$ must explicitly reference and utilize the provided knowledge $D_k$. This design compels any model trained with this paradigm to learn a general reasoning function, $f_{reason}$, where $t \approx f_{\text{reason}}(o, \mathcal{I}, D_k)$, rather than simply memorizing patterns. This learned reasoning ability is crucial for generalization. When the model later encounters an unseen observation, $o_{ood}$, it can apply its reasoning function to new knowledge, $D_{ood}$, to perform effectively in OOD scenarios: $f_{reason}(o_{ood}, \mathcal{I}, D_{ood})$. Using this paradigm, we train our end-to-end native agent, DKR-GUI, with prompt in table 8.

In our implementation, the dynamic knowledge $D_k$ consists of two forms of information: Similar Trajectories $\{d_1, d_2, ..., d_k\}$ and Meta Functions $\{f_m\}$. Similar Trajectories provide concrete, step-by-step operational examples from related tasks, guiding the model's immediate decision-making. Meta Functions are reusable, high-level subtasks abstracted from multiple trajectories, such as *def search_in_google(term: str)*, which aid the model in robust planning. These two knowledge forms are complementary, offering the model both specific examples and general strategies.

## 3.3 MA-DKR: A MODULARLY ENHANCED FRAMEWORK

The MA-DKR framework combines the DKR-GUI with two specialized components: a DKSeeker and an executing agent, as shown in figure 3.

The DKSeeker serves as the framework's dynamic knowledge source. It implements a retrieval function $R : \mathcal{I} \times \mathcal{K} \to D_k$, which identifies relevant knowledge based on the semantic similarity

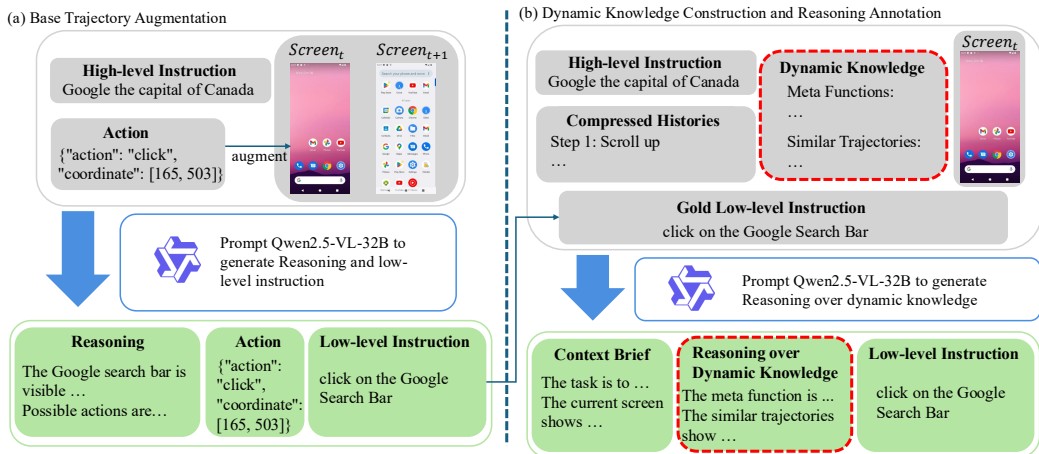

Figure 4: **The two-stage process for generating training data for DKRF.** (a) First, we augment base trajectories by using an MLLM to generate intermediate reasoning and low-level instructions. (b) Second, we construct dynamic knowledge for each sample and then prompt the MLLM to generate a new reasoning process that is explicitly conditioned on this knowledge. The second stage ensures the final training data teaches the model to actively utilize dynamic knowledge for decision.

between instructions, where $\mathcal{K}$ represents the knowledge base containing dynamic knowledge. This process is formalized as:

$$R(i, \mathcal{K}) = \underset{d_j \in \mathcal{K}}{\text{top-k}}\{d_j \mid \text{sim}(\mathcal{I}, i_j) \geq \tau_s\}. \tag{2}$$

where $\mathcal{I}$ is the current instruction, $i_j$ is the instruction associated with knowledge entry $d_j$ in knowledge base, $\text{sim}(\cdot, \cdot)$ is a similarity function, and $\tau_s$ is a similarity threshold.

The executing agent is responsible for the final interaction with the GUI. For this role, we use a SOTA GUI agent with strong visual grounding capabilities. Instead of handling the complex, high-level user instruction, the executing agent focuses on carrying out the simpler, low-level instructions generated by the DKR-GUI. This design creates a complementary structure: DKR-GUI provides robust planning, while the executing agent offers reliable, fine-grained operation.

### 3.4 Training Data Generation

High-quality training data is essential for DKRF. Our data generation process has two steps, as shown in figure 4, with prompt in table 9 and 10.

**Base Trajectory Augmentation.** Since most existing GUI datasets (Rawles et al., 2023; Lu et al., 2024; Zhang et al., 2025b; Cheng et al., 2025) lack intermediate reasoning steps, we first use Qwen2.5-VL-32B (Bai et al., 2025), a model trained on some GUI datasets, to annotate the original trajectories, balancing both performance and cost-efficiency. For each step, it generates a "thought" and a "low-level instruction," creating a base dataset enriched with reasoning (Zhang et al., 2024).

**Dynamic Knowledge Construction and Annotation.** For each sample, we construct specific dynamic knowledge. Based on the semantic similarity of instructions, we retrieve relevant trajectories from the training set to avoid making prior assumptions about the task. This knowledge is then presented in two forms: 1) similar trajectories provides concrete, step-by-step operational examples from past tasks. It focuses on guiding the model's immediate, low-level decision-making. 2) meta functions consist of reusable, code-like functions that represent higher-level subtasks, summarized from multiple similar trajectories. It focuses on providing the model with a more abstract plan.

Finally, to maintain consistency, we use the same Qwen2.5-VL-32B model to generate a thought process for each sample that explicitly references this constructed dynamic knowledge. We extracted 10k trajectories to train DKR-GUI. 50% of these were replaced with data containing dynamic knowledge. In ablation studies, we also systematically compare the impact of different ratios.

Table 1: **Comparison on In-Distribution benchmarks.** Our methods are highlighted in gray and the SOTA results are shown in **bold**. '-' in Planning column signifies an end-to-end model, whereas those specifying DKR-GUI as planning agent represent modular agent frameworks. Average SR is the mean success rate over the two benchmarks.

| Planning | Executing | AITZ | | | Android Control | | | Average |
| | | Type | Grounding | SR | Type | Grounding | SR | SR |
|---|---|---|---|---|---|---|---|---|
| - | GUI-R1-3B | 55.4 | 53.9 | 36.7 | 58.0 | 56.2 | 46.6 | 41.7 |
| - | InfiGUI-3B | 67.0 | 61.9 | 52.4 | 82.7 | 74.4 | 71.7 | 62.1 |
| - | Aguvis-7B | 35.7 | - | 19.0 | 65.6 | - | 54.2 | 36.6 |
| - | UI-TARS-7B | 69.1 | 58.4 | 52.8 | 78.8 | 76.7 | 71.8 | 62.3 |
| - | UI-TARS-1.5 | 72.6 | 60.8 | 56.5 | 75.6 | 73.8 | 67.1 | 61.8 |
| - | OS-Genesis-7B | 20.0 | - | 8.5 | 65.9 | - | 44.4 | 26.4 |
| - | OS-ATLAS-7B | 72.1 | 59.8 | 58.1 | 72.6 | 75.5 | 64.7 | 61.4 |
| - | Qwen2.5-VL-7B | 78.0 | 66.7 | 65.1 | 74.2 | 71.3 | 65.7 | 65.4 |
| GPT-4 | Qwen2.5-VL-7B | 77.3 | 62.9 | 63.0 | 73.5 | 71.9 | 64.9 | 64.0 |
| Claude | Qwen2.5-VL-7B | 76.5 | 64.6 | 63.3 | 73.9 | 69.2 | 65.3 | 64.3 |
| - | DKR-GUI | 78.6 | 63.9 | 65.1 | 77.0 | 69.6 | 68.6 | 66.9 |
| DKR-GUI | Qwen2.5-VL-7B | **81.8** | **67.0** | **68.2** | 77.6 | 69.7 | 69.4 | 68.8 |
| DKR-GUI | UI-TARS-7B | 73.1 | 63.1 | 56.1 | 78.3 | **76.9** | 71.0 | 63.6 |
| DKR-GUI | OS-ATLAS-7B | 80.2 | 63.1 | 65.1 | **82.2** | 72.4 | **72.9** | **69.0** |

## 4 EXPERIMENTS

### 4.1 EXPERIMENTAL SETUP

**Benchmarks**. We evaluate our methods on four GUI benchmarks. For ID evaluation, we use the standard splits of Android Control (AC) (Li et al., 2024a) and AITZ (Zhang et al., 2024). For OOD evaluation, we use CAGUI (Zhang et al., 2025b) and Kairos (Cheng et al., 2025). As these benchmarks were released after our base model's pre-training cutoff, we infer them to be OOD, ensuring a fair assessment of generalization by minimizing the likelihood of data contamination. As our research focuses on the model's reasoning and planning capabilities, the tasks are all high-level.

**Baseline**. We compare our methods against two categories of SOTA baselines. For end-to-end models, we select leading open-source agents like OS-ATLAS-7B (Wu et al., 2024b), UI-TARS-7B (Qin et al., 2025), OS-Genesis-7B (Sun et al., 2024), Aguvis-7B (Xu et al., 2024) as well as our base model, Qwen2.5-VL-7B (Bai et al., 2025). For modular agent frameworks, we construct powerful baselines by combining SOTA planning agents and executing agents (Qwen2.5-VL-7B, UI-TARS-7B, OS-ATLAS-7B).

**Implementation Details**. Our DKR-GUI is fine-tuned from Qwen2.5-VL-7B. The fine-tuning dataset consists of approximately 10k trajectories; to implement our DKRF paradigm without increasing the training cost, half of these are reformatted to include dynamic knowledge, replacing their original versions. All open-source baselines are also 7B models to ensure a fair comparison. In the MA-DKR framework, the default executing agent is Qwen2.5-VL-7B, and the DKSeeker uses the all-MiniLM-L6-v2 (Khashabi et al., 2021; Lewis et al., 2021) model for calculating semantic similarity. We strictly follow the evaluation protocols of existing work (Bai et al., 2025; Zhang et al., 2025b; Wu et al., 2024b) and have reproduced some baseline results to ensure direct comparisons in an identical environment, more details in section A.

### 4.2 MAIN RESULTS

We first evaluate our methods against SOTA agents on both ID and OOD benchmarks, with results in table 1 and table 2.

**DKR-GUI shows strong generalization.** In OOD benchmarks, DKR-GUI achieves an average step success rate (SR) of 70.4%, significantly outperforming other models. This result provides strong evidence that our DKRF paradigm effectively enhances generalization by shifting the model from memorization to reasoning. Importantly, this gain in OOD performance does not come at the cost of ID performance; DKR-GUI (66.9%) also surpasses its baseline (65.4%) on ID tasks.

Table 2: **Comparison on Out-of-Distribution benchmarks.** The setup is the same as in Table 1.

| Planning | Executing | CAGUI | | | Kairos | | | Average |
| | | Type | Grounding | SR | Type | Grounding | SR | SR |
|---|---|---|---|---|---|---|---|---|
| - | GUI-R1-3B | 71.9 | 45.9 | 36.9 | 67.4 | 69.2 | 48.8 | 42.8 |
| - | InfiGUI-3B | 81.2 | 46.2 | 45.9 | 73.0 | 72.6 | 61.8 | 53.9 |
| - | Aguvis-7B | 67.4 | - | 38.2 | - | - | - | - |
| - | UI-TARS-7B | 74.4 | 43.7 | 47.9 | 65.6 | 51.7 | 53.5 | 50.7 |
| - | UI-TARS-1.5 | 88.1 | 54.3 | 58.9 | 87.4 | 76.4 | 74.9 | 66.9 |
| - | OS-Genesis-7B | 38.1 | - | 14.5 | - | - | - | - |
| - | OS-ATLAS-7B | 73.8 | 34.7 | 39.5 | 68.0 | 46.2 | 49.3 | 44.4 |
| - | Qwen2.5-VL-7B | 82.0 | 51.9 | 53.6 | 88.4 | 82.4 | 78.1 | 65.9 |
| GPT-4 | Qwen2.5-VL-7B | 84.5 | 55.5 | 55.9 | 88.0 | 79.6 | 77.5 | 66.7 |
| Claude | Qwen2.5-VL-7B | 82.7 | 53.4 | 53.8 | 85.0 | 76.1 | 73.1 | 63.5 |
| - | DKR-GUI | 87.8 | **57.6** | **61.2** | 88.6 | 82.3 | 79.6 | **70.4** |
| DKR-GUI | Qwen2.5-VL-7B | 86.2 | 55.8 | 58.0 | **90.2** | **85.0** | **81.7** | 69.9 |
| DKR-GUI | UI-TARS-7B | 80.7 | 53.9 | 55.3 | 72.9 | 64.2 | 62.3 | 58.8 |
| DKR-GUI | OS-ATLAS-7B | **89.1** | 51.3 | 57.2 | 89.3 | 71.3 | 71.6 | 64.4 |

Table 3: **Ablation study of DKRF paradigm.** DKR-GUI is evaluated in both end-to-end and modular agent framework settings. In the modular agent framework, different variants of DKR-GUI serve as the planning agent, while the executing agent is Qwen2.5-VL-7B. The vanilla fine-tune variant, which uses no dynamic knowledge during either training or testing, serves as the baseline.

| Similar Traj | Meta Func | Method | In-Distribution | | | Out-of-Distribution | | | Avg. Len. of DK |
| | | | AITZ | AC | Average | CAGUI | Kairos | Average | |
|---|---|---|---|---|---|---|---|---|---|
| | | | *end2end* | | | | | | |
| ✗ | ✗ | vanilla FT | 65.2 | 68.6 | 66.9 | 54.4 | 73.2 | 63.8 | 0 |
| | ✓ | | 64.8 | 68.5 | 66.7 (↓ 0.2%) | 58.9 | 77.7 | 68.3 (↑ 4.5%) | 388 |
| ✓ | | DKR-GUI | 64.8 | 68.7 | 66.8 (↓ 0.0%) | 61.2 | 79.1 | 70.2 (↑ 6.4%) | 533 |
| ✓ | ✓ | | 65.1 | 68.6 | 66.9 (↓ 0.0%) | 61.2 | 79.6 | 70.4 (↑ 6.6%) | 921 |
| | | | *modular agent framework* | | | | | | |
| ✗ | ✗ | vanilla FT | 68.8 | 69.3 | 69.1 | 54.8 | 79.3 | 67.1 | 0 |
| | ✓ | | 68.1 | 69.2 | 68.7 (↓ 0.4%) | 56.8 | 81.0 | 68.9 (↑ 1.8%) | 388 |
| ✓ | | DKR-GUI | 68.5 | 69.3 | 68.9 (↓ 0.2%) | 57.6 | 81.2 | 69.4 (↑ 2.3%) | 533 |
| ✓ | ✓ | | 68.2 | 69.4 | 68.8 (↓ 0.3%) | 58.0 | 81.7 | 69.9 (↑ 2.8%) | 921 |

Table 4: **Ablation study of reasoning.** This study investigates the importance of generating a reasoning process that explicitly references dynamic knowledge. The 'None' variant, which receive dynamic knowledge but is not required to reference it in its reasoning process, serves as the baseline.

| Reasoning Generator | In-Distribution | | | Out-of-Distribution | | |
| | AITZ | AC | Average | CAGUI | Kairos | Average |
|---|---|---|---|---|---|---|
| | *end2end* | | | | | |
| None | 64.4 | 69.1 | 66.8 | 56.7 | 77.2 | 67.0 |
| Qwen2.5-VL-7B | 65.9 | 68.6 | 67.3 (↑ 0.5%) | 58.2 | 77.8 | 68.0 (↑ 1.0%) |
| Qwen2.5-VL-32B | 65.1 | 68.6 | 66.9 (↑ 0.1%) | 61.2 | 79.6 | 70.4 (↑ 3.4%) |
| | *modular agent framework* | | | | | |
| None | 68.2 | 69.6 | 68.9 | 55.7 | 80.3 | 68.0 |
| Qwen2.5-VL-7B | 69.1 | 69.4 | 69.3 (↑ 0.4%) | 57.0 | 80.6 | 68.8 (↑ 0.8%) |
| Qwen2.5-VL-32B | 68.2 | 69.4 | 68.8 (↓ 0.1%) | 58.0 | 81.7 | 69.9 (↑ 1.9%) |

**The MA-DKR framework is highly effective.** MA-DKR achieves SOTA-level performance, especially on ID tasks. When DKR-GUI acts as the planning agent, it consistently improves the performance of different executing agents. This shows that the division of labor is a very effective strategy, where the DKR-GUI provides high-level plans, simplifying the task for the executing agent.

Table 5: **Ablation study of MA-DKR.** This table evaluates various configurations of MA-DKR. For agents capable of independent operation (Qwen2.5-VL and OS-ATLAS), the results show the performance improvement of incorporating them into the MA-DKR framework versus their independent performance. For GUI-Actor, a GUI grounding model which requires a planning agent, the results show the improvement of other agents within the framework, relative to the GPT-4o variant.

| Planning Agent | In-Distribution | | | Out-of-Distribution | | |
|---|---|---|---|---|---|---|
| | AITZ | AC | Average | CAGUI | Kairos | Average |
| *Executing Agent: Qwen2.5-VL-7B* | | | | | | |
| None | 65.1 | 65.7 | 65.4 | 53.6 | 78.1 | 65.9 |
| GPT-4o | 63.0 | 64.9 | 64.0 (↓ 1.4%) | 55.9 | 77.5 | 66.7 (↑ 0.8%) |
| Qwen2.5-VL-7B | 65.8 | 65.1 | 65.5 (↑ 0.1%) | 56.1 | 74.6 | 65.4 (↓ 0.5%) |
| DKR-GUI | 68.2 | 69.4 | 68.8 (↑ 3.4%) | 58.0 | 81.7 | 69.9 (↑ 4.0%) |
| *Executing Agent: OS-ATLAS-7B* | | | | | | |
| None | 58.1 | 64.7 | 61.4 | 39.5 | 49.3 | 44.4 |
| GPT-4o | 53.8 | 62.4 | 58.1 (↓ 3.3%) | 49.2 | 55.9 | 52.6 (↑ 8.2%) |
| Qwen2.5-VL-7B | 57.7 | 66.4 | 62.1 (↑ 0.7%) | 42.4 | 57.8 | 50.1 (↑ 5.7%) |
| DKR-GUI | 65.1 | 72.9 | 69.0 (↑ 7.6%) | 57.2 | 71.6 | 64.4 (↑ 20.0%) |
| *Executing Agent: GUI-Actor-7B* | | | | | | |
| None | - | - | - | - | - | - |
| GPT-4o | 33.2 | 49.7 | 41.5 | 38.9 | 56.9 | 47.9 |
| Qwen2.5-VL-7B | 60.1 | 60.6 | 60.4 (↑ 18.9%) | 61.8 | 76.6 | 69.2 (↑ 21.3%) |
| DKR-GUI | 63.9 | 71.0 | 67.5 (↑ 26.0%) | 61.9 | 78.5 | 70.2 (↑ 22.3%) |

## 4.3 ABLATION STUDIES

Next, we investigate the impact of different components on our method's performance. Our default modular agent framework configuration consists of DKR-GUI as the planning agent and Qwen2.5-VL-7B as the executing agent. More detailed results for ablation studies are provided in the appendix C.

**The DKRT paradigm is crucial for overcoming the limitations of vanilla fine-tuning, as shown in Table 3, 12.** While vanilla fine-tuning often improves ID performance at the cost of OOD generalization by "memorizing" in-distribution patterns, DKRT fosters "dynamic reasoning", as DKR-GUI's success rate increased by 6.6%, and MA-DKR's by 2.8%, all without degrading ID performance. These findings confirm that DKRT effectively addressing the overfitting and poor generalization issues inherent in vanilla fine-tuning approaches.

**The components of dynamic knowledge are functionally complementary, as shown in Table 3, 12.** For both end-to-end and agent framework settings, "Similar Trajectories" generally provide a more direct performance boost. In contrast, "Meta Functions" contribute by teaching the model a more generalized planning approach, abstracting operations into higher-level intents (e.g., *search_in_google*). It provides essential structural compression capabilities in efficiency-sensitive real-world scenarios. Thus, the two components work in synergy.

**The intermediate reasoning process is indispensable for effective knowledge utilization, as shown in Table 4, 13.** Removing the explicit reasoning step significantly hinders the model's ability to leverage dynamic knowledge, degrading OOD performance by forcing a reliance on brittle pattern-matching. Furthermore, the quality of the reasoning process directly impacts agent effectiveness. Models trained on reasoning chains generated by the more capable Qwen2.5-VL-32B achieve superior OOD results compared to those trained with data from the 7B model, confirming that higher-quality reasoning serves as a more effective supervisory signal.

**The MA-DKR framework universally enhances the performance of various executing agents, as shown in Table 5.** Decoupling tasks into planning and executing significantly boosts the performance of single GUI agents across both ID and OOD scenarios. Besides, we find GUI-Actor struggles with navigation even when paired with GPT-4o as a SOTA GUI grounding model. To some degree, this highlights the limitations of GUI Grounding models. When guided by a capable

planning agent, its effectiveness dramatically increases to 67.5% and 70.0%. This synergy allows the MA-DKR pair to even surpass the performance of the independent DKR-GUI.

**DKR-GUI excels as planning agent, boosting executing agent performance across multiple benchmarks, as shown in Table 5.** Its superiority over its base model, Qwen2.5-VL, when used as planning agent, directly validates the effectiveness of the DKRF paradigm. Notably, the initially weaker OS-ATLAS, when guided by DKR-GUI, surpasses variants using the more capable Qwen2.5-VL as executing agent. This highlights that an agent's instruction-following ability is a critical research area, separate from its autonomous capabilities.

**DKRF achieves significant gains with only a small increase in cost, as shown in Table 7, Table 6.** We present a cost analysis of DKRF regarding data, training, and inference. As shown, DKRF uses a relatively small training set (10k trajectories). The annotation cost for building the training data is also acceptable. During inference, the cost increases by only 10% to 20% compared to vanilla SFT

Table 6: Statistics of Datasets and Annotation Details

| Dataset | Tasks Overall | Screenshots | Base Trajectory Augmentation | Dynamic Knowledge Construction and Reasoning Annotation | |
|---|---|---|---|---|---|
| | | | Input Tokens Pre Step | Tasks w/ dynamic knowledge | Input Tokens Pre Step (w/ DK) |
| AITZ | 1.0K | 1.0K | 2.3k | 0.5K | 2.3K |
| Android Control | 6.0K | 35.4K | 7.3k | 3K | 3.9K |
| GUI Odyssey | 3.0K | 46.2K | 11.0k | 1.5K | 4.3K |

Table 7: Comparison of Computational Cost during Inference

| Method | FLOPs | Input Tokens Pre Step |
|---|---|---|
| Vanilla SFT | 104.2 | 3.6K |
| DKRF | 118.8 | 4.4K |

## 5   CONCLUSION

To improve the OOD generalization of mobile GUI agents, we introduce DKRT paradigm that shifts the learning objective from pattern memorization to knowledge-conditioned reasoning. DKR-GUI, a robust end-to-end native mobile agent, establishes a new SOTA for open-source models on OOD mobile benchmarks. When integrated into MA-DKR modular framework, it also achieves SOTA performance on ID mobile tasks through an effective division of labor. This work marks a significant step in evolving mobile GUI agents from brittle pattern-matchers to robust dynamic reasoners, opening avenues for developing self-evolving agents capable of lifelong learning.

**Ethics Statement.** This research adheres to the ICLR Code of Ethics. Our work is built upon publicly available datasets, and the data generation process was conducted in a controlled, offline environment. No private user data or human subjects were involved in our experiments. The primary goal of this research is to advance the capabilities of GUI agents for positive applications, such as improving productivity and accessibility.

**Reproducibility Statement.** We are committed to ensuring the reproducibility of our research. All experiments were conducted on publicly available benchmarks: Android Control (AC), AITZ, CAGUI, and Kairos. Our methodology, including DKRF paradigm, DKR-GUI and the MA-DKR framework, is detailed in section 3. The complete process for training data generation is described in section 3.4 and illustrated in figure 4, with the specific prompts provided in section B in the Appendix. Key implementation details, including the base models (Qwen2.5-VL-7B), hyperparameters, and evaluation protocols, are provided in section 4.1. Upon acceptance, we will release our source code, generated training data, and fine-tuned model weights to facilitate verification of our results and further research by the community.

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

# A DETAILS OF EVALUATION

**Benchmarks.** The Android Control (Li et al., 2024a) benchmark provides both low-level and high-level evaluation settings. The former employs a mix of low- and high-level instructions to assess single-step execution capabilities. In contrast, the latter uses exclusively high-level instructions, designed to evaluate the model's high-dimensional reasoning and planning abilities. Our evaluations across multiple benchmarks are based on the high-level, pure-vision setting to systematically investigate the model's reasoning and planning performance in both ID and OOD scenarios.

**Metrics.** To ensure a fair comparison across all baseline methods, we standardize the evaluation metrics for each action, consistent with OS-ATLAS (Wu et al., 2024b). For click-based actions (e.g., CLICK, LONG_PRESS), the model generate both the action type and the position coordinates (x, y). Since ground-truth bounding boxes are not always available in the test data, we measure performance by calculating the distance between the predicted and ground-truth coordinates. Following (Lu et al., 2024), a coordinate is considered correct if it falls within a distance of 14% of the screen width from the ground-truth location. For input-based actions (e.g., TYPE, OPEN_APP), a step is considered correct if and only if both the action type and its content are correct. We calculate the F1 score between the predicted and ground-truth text; the text is deemed correct if the F1 score is greater than 0.5. For scroll actions, the directional parameter (i.e., up, down, left, or right) must exactly match the ground-truth. For other actions (e.g., PRESS_BACK), they must be an exact match with the ground-truth to be considered correct.

We evaluate our models using three key metrics common in GUI agent research (Wu et al., 2024b; Zhang et al., 2025b). Type measures the exact match accuracy of the predicted action type (e.g., CLICK, SCROLL). Grounding assesses the precision of coordinate-based actions (e.g., CLICK, LONG_PRESS). SR (Step Success Rate) is the most stringent metric, requiring both the action type and its corresponding arguments (like coordinates) to be correct for a step to be considered successful.

# B PROMPTS

Table 8: Prompt used for DKR-GUI, unified across multiple benchmarks. Due to context length constraints, the action space is encoded into the model's parameters during fine-tune.

**System Message**

You are a helpful AI mobile phone operating assistant.
You need to analyze the user's instructions to identify the actions
that can be performed on the current page, as well as the gui elements associated with those actions.
Possible operations include click, scroll, type, press back,
press home, press enter, long press, open app, wait, task complete, task impossible, etc.

There are also prior functions regarding the extracted actions from
similar instruction trajectories, which can be used as references though not entirely accurate:
{Meta Function}

Several similar trajectory for reference is as follows:
{Similar Trajectories}

The gui elements should be accessible in current page.
The screen's resolution is {width}x{height}.

**User Message**

The user query: {instruction}.

Firstly, evaluate the task completion status based on past actions.
Then, explain your reasoning step-by-step in <thinking></thinking>tags.
Next, summarize your action in <conclusion></conclusion>tags.
Finally, output the specific actions in JSON format in <tool_call></tool_call>tags.

Task Progress (You have done the following operation on the current device):
{History}

Table 9: Prompt used for Base Trajectory Augmentation.

**Role Definition**

You're an expert in mobile app UI navigation.

Given a goal: {instruction}, a screenshot and the action: {action} based on current status,
if there are two image, the second image is the next status.

**Action Space**

'action_type' is one of [
    TYPE: type text in the activated area,
    CLICK: click on a specific position,
    SCROLL: swipe operation,
    PRESS_BACK: return to previous screen,
    PRESS_HOME: return to home screen,
    PRESS_ENTER: press the enter key,
    STATUS_TASK_COMPLETE: task is completed,
    STATUS_TASK_IMPOSSIBLE: task is not completed within limited steps,
    LONG_PRESS: click on a specific position and hold few seconds,
    OPEN_APP: open an app,
    WAIT: wait for the current page loading.
]

'action_text' means the input text if a input area is activated,
'touch_xy' is the touch position, formated as [x, y].
Please note that actions involving click operations (CLICK or LONG_PRESS)
are indicated by red circles in the images to assist perception.

**Task Guidance**

Please analyze the screenshot and **output as following format** in concise words:
1. Summarize the action, for example,
    if the action is to click on a Google search bar
    which is located at the middle and lower part of the screen according to the click position,
    the summary is 'click on the Google search bar located at the middle and lower part of the screen.';
    If the action is 'scroll up', the summary is 'scroll up';
    If the action is 'STATUS_TASK_COMPLETE',
    the summary can be 'stop and set the query as completed'.
    The action summarization must be short and accurate, and **must be wrapped in <action></action>**

2. Under the current page, please analyse the possible actions to take for achieving the goal,
    remember, do not analyse the given action, thinking must limited in 100 words,
    and the answer **must be wrapped in <thinking></thinking>**

3. By doing so, what will happen next, if there are two images,
    you can refer to the second image, the result **must be wrapped in <result></result>**

**Output Format**

Remember, you must answer based on the provided action:
{action}, answers of the above three tasks must be wrapped in corresponding symbols.
When action type is 'STATUS_TASK_COMPLETE' or 'STATUS_TASK_IMPOSSIBLE',
you can not answer like 'Since the action type is STATUS_TASK_COMPLETE',
and do not use 'the user' in the above tasks' response.

Table 10: Prompt used for Dynamic Knowledge Construction and Annotation.

**Role Definition**

You are a reasoning module capable of reconstructing the thought process
that leads from the given premises to the result (gold_low_level_instruction),
utilizing the GUI Agent context.

**Input Instructions**

Given:
    human_instruction - user's high-level command
    history_low_level_instructions - chronological list of past low-level instructions
    meta_functions - prior functions regarding the extracted actions from similar instruction trajectories,
                        which can be used as references though not entirely accurate.
                        (Python-style pseudocode)
    similar_trajectories - trajectories whose instruction are similar to the current trajectory,
                        and be related to meta_functions.
    gold_low_level_instruction - the single low-level instruction that should be executed next
                                (**DO NOT cite during reasoning**)
    current_screen - the raw screenshot

**Task Guidance**

1. **Never mention gold_low_level_action as evidence duiring reasoning**
2. Use **concise**, formal English inside all JSON values.
3. No extra keys, comments, or line breaks outside the JSON object.
4. Never write the word "JSON" or these rules inside any value.

**Output Format**

Return valid JSON with exactly:
{{
"context_brief": string, // ≤ **2 sentences**, combine:
                        // **Never mention gold_low_level_action as evidence duiring reasoning**
                        // (a) progress status
                        // (b) key GUI cues from current_screen
"relevant_meta_thought": string, // concise reasoning on meta_functions and similar_trajectories:
                        // **Never mention gold_low_level_action as evidence duiring reasoning**
                        // Please identify meta functions and similar_trajectories
                        // that are useful to get the gold_low_level_instruction,
                        // and explain how the contents lead to the gold_low_level_instruction.
                        // If none of the meta functions is aligned with the gold_low_level_instruction,
you have to explain the gold_low_level_instruction only based on your own knowledge.
"derived_low_level_instruction": string // identical to gold_low_level_action
}}

**Input**

### Human Instruction:
{instruction}

### History Low-Level Instruction:
{historys}

### Meta Functions Library:
{meta_functions}

### Similar Trajectories:
{similar_trajectories}

### Target Low-Level Instruction (ground truth):
{step_instruction}

### Your job
Follow the System instructions and return the required **JSON**.

# C MORE ABLATIONS

To complement the ablation studies, we conduct additional experiments with different executing agents. We also investigate the impact of the fine-tuning data ratio, the generalizability of the DKRF paradigm across different MLLMs, and the effect of dynamic knowledge relevance. These experiments reaffirm the conclusions drawn in the main text and provide the following insights.

**Dynamic knowledge activates the latent reasoning capabilities of (M)LLMs, as shown in Table 11.** We demonstrate that LLM's reasoning capabilities can be activated by external knowledge without fine-tuning. Providing dynamic knowledge to a modular agent framework solely through in-context prompting improves its OOD performance from 65.4% to 67.9% (+2.5%), while ID performance remains stable. This result indicates that leveraging dynamic knowledge is a viable and effective strategy for enhancing OOD generalization. However, a significant challenge remains in how to preserve and augment this capability while simultaneously leveraging fine-tuning to enhance agent's GUI comprehension.

**The ratio of data with dynamic knowledge in training set impacts model performance, as shown in Table 14.** We observe that a higher proportion of data "w/o dynamic knowledge" tends to improve performance on ID benchmarks. However, further increasing the proportion of data "w/ dynamic knowledge" does not yield continuous improvement on OOD benchmarks. We hypothesize this is because data "w/ dynamic knowledge" endows the model with the ability to reason, while data "w/o dynamic knowledge" cultivates its foundational capacity to understand GUI scenarios. An effective balance of both is crucial for achieving optimal performance across all benchmarks.

**The DKRF paradigm generalizes effectively across different MLLMs, as shown in Table 15.** We find that DKRF is also effective when applied to LLaVA (Liu et al., 2023), yielding even greater performance gains than on Qwen2.5-VL. We attribute this to the fact that LLaVA's original training set contains less GUI-specific data. This finding further underscores the efficacy of DKRF in equipping MLLMs with essential knowledge for GUI-based tasks.

**Less relevant dynamic knowledge is less effective but still provides a benefit, as shown in Table 16.** By adjusting the DKSeeker's threshold, we control the relevance of the retrieved dynamic knowledge. As the results indicate, higher relevance correlates with greater performance gains on OOD benchmarks, confirming the importance of valuable knowledge. At the same time, this also demonstrates the robustness of the DKRF paradigm; even when highly relevant examples are unavailable, the model still gains a notable advantage compared to operating without any dynamic knowledge.

**DKRF improves the utilization of dynamic knowledge, as shown in Figure 5.** We compared the vanilla Qwen2.5-VL-7B with models fine-tuned by different methods. We found that DKRF explicitly invokes dynamic knowledge more frequently.

**DKRF remains robust even without dynamic knowledge, as shown in Table 17.** Results show that DKRF maintains performance comparable to vanilla SFT on OOD datasets. This indicates: The model maintains robust performance under conditions without dynamic knowledge; The gains primarily stem from the learning of "knowledge reasoning capability," rather than reliance on retrieval semantic similarity.

Table 11: **Ablation study on the effect of dynamic knowledge in in-context learning.** This experiment is conducted within a modular agent framework where both the planning agents are Qwen2.5-VL-7B, to control for the effects of prompt length. 'Similar Traj' and 'Meta Func' indicate that the dynamic knowledge contains **similar trajectories** and **meta functions**, respectively. The 'Average' column shows the relative performance change of each variant against the baseline (the first row) which uses no dynamic knowledge.

| Similar Traj | Meta Func | Executing Agent | ID | | | OOD | | |
|---|---|---|---|---|---|---|---|---|
| | | | AITZ | AC | Average | CAGUI | Kairos | Average |
| ✗ | ✗ | | 65.8 | 65.1 | 65.5 | 56.1 | 74.6 | 65.4 |
| | ✓ | Qwen2.5-VL | 66.0 | 65.0 | 65.5 (↓ 0.0%) | 56.4 | 78.6 | 67.5 (↑ 2.1%) |
| ✓ | | | 65.3 | 65.2 | 65.3 (↓ 0.2%) | 56.1 | 78.1 | 67.1 (↑ 1.7%) |
| ✓ | ✓ | | 65.6 | 65.5 | 65.6 (↑ 0.1%) | 56.6 | 79.1 | 67.9 (↑ 2.5%) |
| ✗ | ✗ | | 55.1 | 68.2 | 61.7 | 48.9 | 55.9 | 52.4 |
| | ✓ | UI-TARS | 54.7 | 69.0 | 61.9 (↑ 0.2%) | 50.8 | 58.9 | 54.9 (↑ 2.5%) |
| ✓ | | | 54.9 | 68.6 | 61.8 (↑ 0.1%) | 50.8 | 59.4 | 55.1 (↑ 2.7%) |
| ✓ | ✓ | | 55.0 | 69.3 | 62.2 (↑ 0.5%) | 53.2 | 62.1 | 57.7 (↑ 5.3%) |
| ✗ | ✗ | | 57.7 | 66.4 | 62.1 | 42.4 | 57.8 | 50.1 |
| | ✓ | OS-ATLAS | 59.0 | 67.0 | 63.0 (↑ 0.9%) | 47.9 | 63.3 | 55.6 (↑ 5.5%) |
| ✓ | | | 57.2 | 65.4 | 61.3 (↓ 0.8%) | 47.5 | 64.5 | 56.0 (↑ 5.9%) |
| ✓ | ✓ | | 57.5 | 66.7 | 62.1 (↑ 0.0%) | 47.6 | 64.5 | 56.1 (↑ 6.0%) |

Table 12: **Ablation study of DKRF paradigm.** DKR-GUI is evaluated in both end-to-end and modular agent framework settings. In the modular agent framework, different variants of DKR-GUI serve as the planning agent. The vanilla fine-tune variant, which uses no dynamic knowledge during either training or testing, serves as the baseline

| Similar Traj | Meta Func | Planning Agent | ID | | | OOD | | |
|---|---|---|---|---|---|---|---|---|
| | | | AITZ | AC | Average | CAGUI | Kairos | Average |
| | | | *end2end* | | | | | |
| ✗ | ✗ | vanilla FT | 65.2 | 68.6 | 66.9 | 54.4 | 73.2 | 63.8 |
| | ✓ | | 64.8 | 68.5 | 66.7 (↓ 0.2%) | 58.9 | 77.7 | 68.3 (↑ 4.5%) |
| ✓ | | DKR-GUI | 64.8 | 68.7 | 66.8 (↓ 0.1%) | 61.2 | 79.1 | 70.2 (↑ 6.4%) |
| ✓ | ✓ | | 65.1 | 68.6 | 66.9 (↓ 0.0%) | 61.2 | 79.6 | 70.4 (↑ 6.6%) |
| | | | *Executing Agent: Qwen2.5-VL-7B* | | | | | |
| ✗ | ✗ | vanilla FT | 68.8 | 69.3 | 69.1 | 54.8 | 79.3 | 67.1 |
| | ✓ | | 68.1 | 69.2 | 68.7 (↓ 0.4%) | 56.8 | 81.0 | 68.9 (↑ 1.8%) |
| ✓ | | DKR-GUI | 68.5 | 69.3 | 68.9 (↓ 0.2%) | 57.6 | 81.2 | 69.4 (↑ 2.3%) |
| ✓ | ✓ | | 68.2 | 69.4 | 68.8 (↓ 0.3%) | 58.0 | 81.7 | 69.9 (↑ 2.8%) |
| | | | *Executing Agent: UI-TARS-7B* | | | | | |
| ✗ | ✗ | vanilla FT | 55.8 | 71.0 | 63.4 | 53.9 | 60.9 | 57.4 |
| | ✓ | | 56.2 | 70.8 | 63.5 (↑ 0.1%) | 54.4 | 60.9 | 57.7 (↑ 0.3%) |
| ✓ | | DKR-GUI | 55.8 | 71.1 | 63.5 (↑ 0.1%) | 56.0 | 63.5 | 59.8 (↑ 2.4%) |
| ✓ | ✓ | | 56.1 | 71.0 | 63.6 (↑ 0.2%) | 55.3 | 62.3 | 58.8 (↑ 1.4%) |
| | | | *Executing Agent: OS-ATLAS-7B* | | | | | |
| ✗ | ✗ | vanilla FT | 65.8 | 73.9 | 69.9 | 54.6 | 72.2 | 63.4 |
| | ✓ | | 65.8 | 73.1 | 69.5 (↓ 0.4%) | 55.6 | 70.4 | 63.0 (↓ 0.4%) |
| ✓ | | DKR-GUI | 65.7 | 73.4 | 69.6 (↓ 0.3%) | 57.4 | 71.8 | 64.6 (↑ 1.2%) |
| ✓ | ✓ | | 65.1 | 72.9 | 69.0 (↓ 0.9%) | 57.2 | 71.6 | 64.4 (↑ 1.0%) |

Table 13: **Ablation study of the intermediate reasoning.** This study investigates the importance of generating a thought process that explicitly references dynamic knowledge. The 'None' variant, which is provided with dynamic knowledge but is not required to reference it in its thought process, serves as the baseline.

| Reasoning Generator | ID | | | | OOD | |
| | AITZ | AC | Average | CAGUI | Kairos | Average |
|---|---|---|---|---|---|---|
| | | | *end2end* | | | |
| None | 64.4 | 69.1 | 66.8 | 56.7 | 77.2 | 67.0 |
| Qwen2.5-VL-7B | 65.9 | 68.6 | 67.3 (↑ 0.5%) | 58.2 | 77.8 | 68.0 (↑ 1.0%) |
| Qwen2.5-VL-32B | 65.1 | 68.6 | 66.9 (↑ 0.1%) | 61.2 | 79.6 | 70.4 (↑ 3.4%) |
| | | | *Executing Agent: Qwen2.5-VL-7B* | | | |
| None | 68.2 | 69.6 | 68.9 | 55.7 | 80.3 | 68.0 |
| Qwen2.5-VL-7B | 69.1 | 69.4 | 69.3 (↑ 0.4%) | 57.0 | 80.6 | 68.8 (↑ 0.8%) |
| Qwen2.5-VL-32B | 68.2 | 69.4 | 68.8 (↓ 0.1%) | 58.0 | 81.7 | 69.9 (↑ 1.9%) |
| | | | *Executing Agent: UI-TARS-7B* | | | |
| None | 55.9 | 70.9 | 63.4 | 54.9 | 62.0 | 58.5 |
| Qwen2.5-VL-7B | 56.3 | 70.9 | 63.6 (↑ 0.2%) | 55.7 | 62.8 | 59.3 (↑ 0.8%) |
| Qwen2.5-VL-32B | 56.1 | 71.0 | 63.6 (↑ 0.2%) | 55.3 | 62.3 | 58.8 (↑ 0.3%) |
| | | | *Executing Agent: OS-ATLAS-7B* | | | |
| None | 65.3 | 73.3 | 69.3 | 55.1 | 72.7 | 63.9 |
| Qwen2.5-VL-7B | 65.9 | 73.2 | 69.6(↑ 0.3%) | 55.8 | 69.6 | 62.7 (↓ 1.2%) |
| Qwen2.5-VL-32B | 65.1 | 72.9 | 69.0 (↓ 0.3%) | 57.2 | 71.6 | 64.4 (↑ 0.5%) |

Table 14: **Ablation study on the proportion of training data with dynamic knowledge.** "w/" denotes the proportion of samples containing dynamic knowledge, showing how performance varies with the changing ratio of these two data types.

| w/ | ID | | | OOD | | |
| | AITZ | AC | Average | CAGUI | Kairos | Average |
|---|---|---|---|---|---|---|
| | | | *end2end* | | | |
| 20% | 65.7 | 68.6 | **67.2** | 59.4 | 79.6 | 69.5 |
| 50% | 65.1 | 68.6 | 66.9 | 61.2 | 79.6 | **70.4** |
| 80% | 64.4 | 67.2 | 65.8 | 60.3 | 79.1 | 69.7 |
| | | | *Executing Agent: Qwen2.5-VL-7B* | | | |
| 20% | 68.7 | 69.6 | **69.2** | 56.9 | 82.3 | 69.6 |
| 50% | 68.2 | 69.4 | 68.8 | 58.0 | 81.7 | **69.9** |
| 80% | 67.8 | 68.3 | 68.1 | 57.4 | 81.4 | 69.4 |
| | | | *Executing Agent: UI-TARS-7B* | | | |
| 20% | 56.0 | 71.1 | **63.6** | 55.2 | 63.3 | **59.3** |
| 50% | 56.1 | 71.0 | **63.6** | 55.3 | 62.3 | 58.8 |
| 80% | 55.9 | 70.9 | 63.4 | 55.3 | 62.7 | 59.0 |
| | | | *Executing Agent: OS-ATLAS-7B* | | | |
| 20% | 65.9 | 73.4 | **69.7** | 55.8 | 70.9 | 63.4 |
| 50% | 65.1 | 72.9 | 69.0 | 57.2 | 71.6 | **64.4** |
| 80% | 65.6 | 72.1 | 68.9 | 56.5 | 71.4 | 64.0 |

Table 15: **Ablation study of the DKRF paradigm on the LLaVA model.** We fine-tuned LLaVA using the same data and methodology and tested its performance across multiple benchmarks.

| Method | ID | | | OOD | | |
|---|---|---|---|---|---|---|
| | AITZ | AC | Average | CAGUI | Kairos | Average |
| *end2end* | | | | | | |
| vanilla FT | 41.7 | 49.2 | 45.5 | 23.4 | 32.4 | 27.9 |
| DKRF | 44.9 | 51.6 | 48.3 (↑ 2.8%) | 32.8 | 48.4 | 40.6 (↑ 12.7%) |
| *Executing Agent: Qwen2.5-VL-7B* | | | | | | |
| vanilla FT | 62.7 | 62.8 | 62.8 | 49.5 | 72.4 | 61.0 |
| DKRF | 64.2 | 62.6 | 63.4 (↑ 0.6%) | 54.6 | 76.8 | 65.7 (↑ 4.7%) |
| *Executing Agent: UI-TARS-7B* | | | | | | |
| vanilla FT | 52.8 | 70.4 | 61.6 | 48.4 | 54.3 | 51.4 |
| DKRF | 53.3 | 70.3 | 61.8 (↑ 0.2%) | 51.1 | 60.6 | 55.9 (↑ 4.5%) |
| *Executing Agent: OS-ATLAS-7B* | | | | | | |
| vanilla FT | 58.0 | 67.6 | 62.8 | 38.1 | 42.6 | 40.4 |
| DKRF | 58.1 | 67.6 | 62.9 (↑ 0.1%) | 48.5 | 57.6 | 53.1 (↑ 13.1%) |

Table 16: **Ablation study on the relevance of dynamic knowledge.** We use DKSeeker to retrieve dynamic knowledge based on instruction similarity, with the relevance controlled by a threshold, as Eq. 2. "Threshold" represents different similarity thresholds, where 0.6 is the default setting used in our experiments.

| Threshold | ID | | | ood | | |
|---|---|---|---|---|---|---|
| | AITZ | AC | Average | CAGUI | Kairos | Average |
| *end2end* | | | | | | |
| vanilla FT | 65.2 | 68.6 | 66.9 | 54.4 | 73.2 | 63.8 |
| 0.0 | 65.3 | 68.2 | 66.8 (↓ 0.1%) | 58.5 | 75.4 | 67.0 (↑ 3.2%) |
| 0.3 | 64.1 | 68.3 | 66.2 (↓ 0.7%) | 59.8 | 78.5 | 69.2 (↑ 5.4%) |
| 0.6 | 65.1 | 68.6 | 66.9 (↓ 0.0%) | 61.2 | 79.6 | 70.4 (↑ 6.6%) |
| *Executing Agent: Qwen2.5-VL-7B* | | | | | | |
| vanilla FT | 68.8 | 69.3 | 69.1 | 54.8 | 79.3 | 67.1 |
| 0.0 | 68.3 | 69.1 | 68.7 (↓ 0.4%) | 56.9 | 78.7 | 67.8 (↑ 0.7%) |
| 0.3 | 67.9 | 69.1 | 68.5 (↓ 0.6%) | 57.2 | 80.8 | 69.0 (↑ 1.9%) |
| 0.6 | 68.2 | 69.4 | 68.8 (↓ 0.3%) | 58.0 | 81.7 | 69.9 (↑ 2.8%) |
| *Executing Agent: UI-TARS-7B* | | | | | | |
| vanilla FT | 55.8 | 71.0 | 63.4 | 53.9 | 60.9 | 57.4 |
| 0.0 | 55.9 | 70.7 | 63.3 (↓ 0.1%) | 54.8 | 63.1 | 59.0 (↑ 1.6%) |
| 0.3 | 55.9 | 70.7 | 63.3 (↓ 0.1%) | 54.5 | 61.3 | 57.9 (↑ 0.5%) |
| 0.6 | 56.1 | 71.0 | 63.6 (↑ 0.2%) | 55.3 | 62.3 | 58.8 (↑ 1.4%) |
| *Executing Agent: OS-ATLAS-7B* | | | | | | |
| vanilla FT | 65.8 | 73.9 | 69.9 | 54.6 | 72.2 | 63.4 |
| 0.0 | 65.9 | 73.2 | 69.6 (↓ 0.3%) | 55.9 | 68.8 | 62.4 (↓ 1.0%) |
| 0.3 | 65.2 | 72.5 | 68.9 (↓ 1.0%) | 56.3 | 70.1 | 63.2 (↓ 0.2%) |
| 0.6 | 65.1 | 72.9 | 69.0 (↓ 0.9%) | 57.2 | 71.6 | 64.4 (↑ 1.0%) |

Table 17: Performance comparison of Baseline and DKRF methods.

| Baseline | FT Method | CAGUI | Kairos | Average |
|---|---|---|---|---|
| Qwen2.5-VL-7B | Vanilla SFT | 54.4 | 73.2 | 63.8 |
| | DKRF | 54.0 | 71.5 | 62.8 |

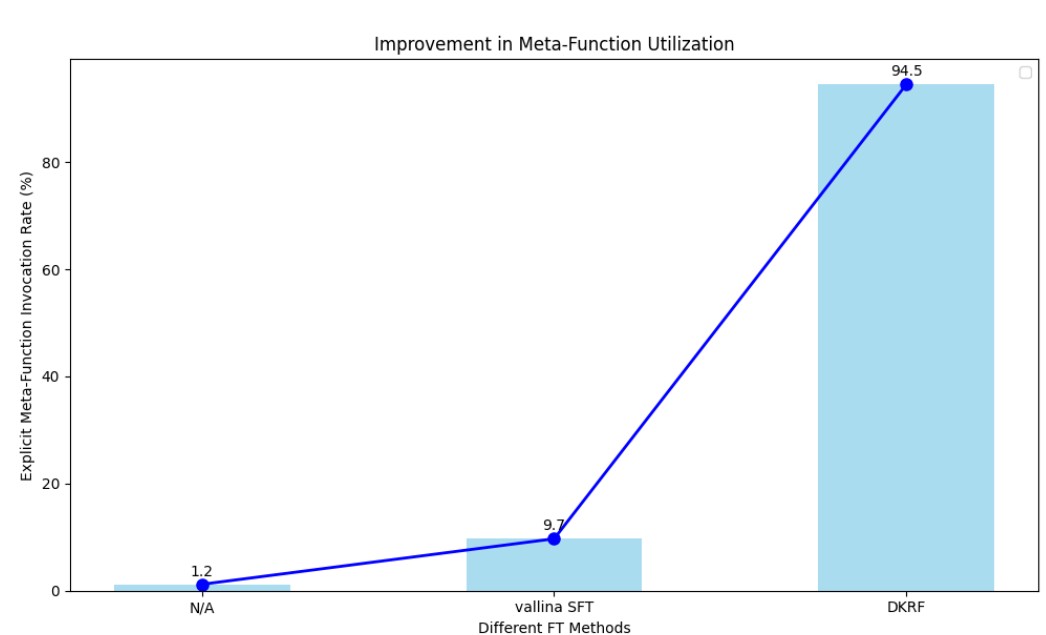

Figure 5: Frequency of dynamic knowledge usage by different methods

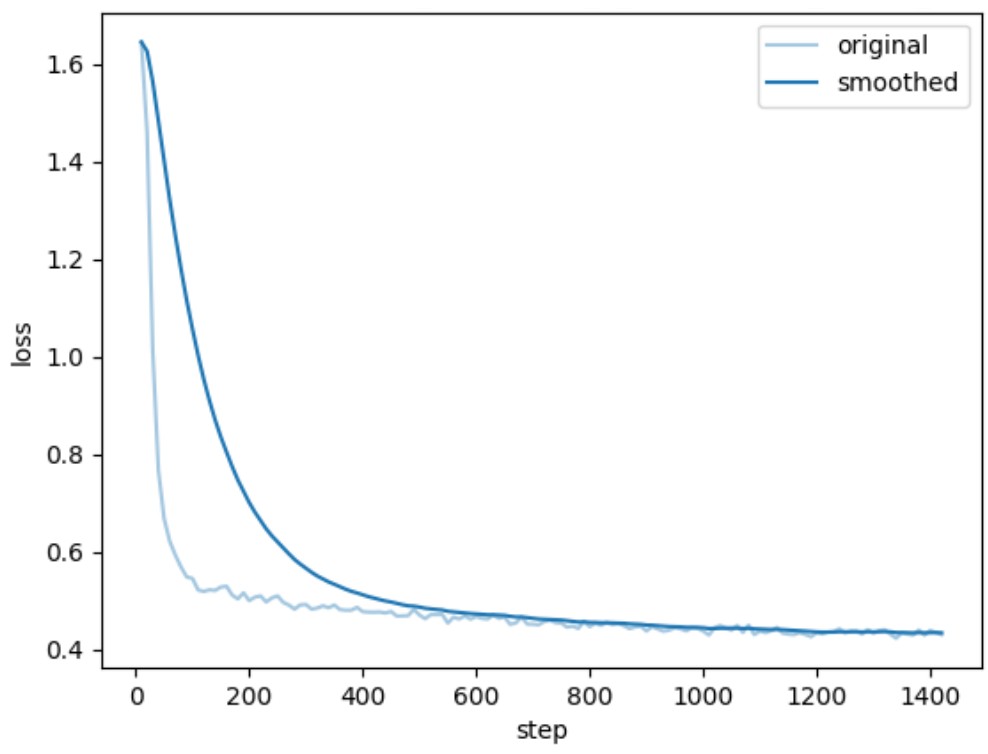

Figure 6: Training Loss Curve of DKRF

## D    MORE RELATED WORK

**RAG in Mobile GUI Agents.**    To address generalization challenges in long-tailed and out-of-distribution GUI tasks, Retrieval-Augmented Generation (RAG) and modular designs have emerged as pivotal paradigms. RAG-GUI (Xu et al., 2025) leverages web tutorials as a non-parametric knowledge base to dynamically generate task-aware guidance during inference. MobileRAG (Park et al., 2025) introduces a comprehensive framework integrating external web knowledge, local application data, and historical operational memory to bolster interaction and self-correction. Regarding knowledge base construction, AppAgent v2 (Li et al., 2024b) documents UI functionalities during an exploration phase for efficient retrieval and updates. LearnAct (Liu et al., 2025a) employs a demonstration-based learning paradigm, utilizing a KnowSeeker agent to retrieve relevant few-shot demonstrations for task guidance. Complementing these retrieval strategies, CoAct-1 (Song et al., 2025) expands the action space via a multi-agent architecture, dynamically delegating tasks to a Programmer or GUI Operator and leveraging code execution to bypass brittle GUI interactions. Collectively, these approaches advance agents from standalone models to compound systems reliant on external knowledge and specialized tools.

However, these approaches predominantly prioritize the optimization of the external retrieval system, focusing on the construction, organization, and updating of knowledge bases. In contrast, DKRF concentrates on enhancing the intrinsic capabilities of the MLLM itself. By fine-tuning the model to effectively comprehend and utilize external dynamic knowledge, DKRF shifts the focus from optimizing retrieval architectures to cultivating the model's core reasoning proficiency.

# E  CASE STUDY

| Meta Functions | Similar Trajectories |
|---|---|
| ```def search_item(search_term: str):```
   ```"""Search for something in the search bar.```
   ```Args:```
      ```search_term (str): The term to search for.```
   ```Returns:```
      ```None: It executes a sequence of actions to perform the search.```
   ```Examples:```
      ```search_item("switch accessories")```
   ```"""```
   ```click on the search_bar```
   ```type in the content: search_term```
   ```press enter | click on a search suggestion``` | **High-level Instruction**: What's the latest video from GameSpot Reviews?
**- Step 1**: click on the search bar located at the bottom middle.
- Step 2: type in the content: "What's the latest video from GameSpot Reviews?"
**- Step 3**: click on the first search suggestion '... reviews 2022' located just below the search bar.
**- Step 4**: stop and set the query as completed.

**High-level Instruction**: What's a good restaurant in Miami?
**- Step 1**: press the home button.
**- Step 2**: click on the Google Search bar located at the bottom right.
**- Step 3**: click on the search bar located at the top right.
**- Step 4**: type in the content: "What's a good restaurant in Miami?"
**- Step 5**: press enter.
**- Step 6**: stop and set the query as completed. |

| Rewritten Trajectories |
|---|
| **High-level Instruction**: What's the latest video from GameSpot Reviews?
**- Step 1**: search_item("What's the latest video from GameSpot Reviews?")
**- Step 2**: task complete |
| **High-level Instruction**: What's a good restaurant in Miami?
**- Step 1**: press home
**- Step 2**: search_item("What's a good restaurant in Miami?")
**- Step 3**: task complete |

Figure 7:  Example of meta-function annotation.

| Meta Functions | Similar Trajectories |
|---|---|
| ```def rename_item(new_name: str):```
    """Rename an existing file or folder in a GUI interface.
    Args:
        new_name (str): The new name to assign to the item.
    Returns:
        None: Executes click, type, and confirmation actions to rename the item.
    Examples:
        rename_item("Project Plan 2024")
    """
    click on the Rename option
    # Optional: click delete/backspace to clear old text if needed
    type in the content: new_name
    click on the Rename button

```def move_item_to_folder(folder_name: str):```
    """Move a selected item to a specified folder.
    Args:
        folder_name (str): The name of the target folder to move the item into.
    Returns:
        None: Moves the current selected item to the given folder.
    Examples:
        move_item_to_folder("Archive 2023")
    """
    click on the Move option
    click on the folder named folder_name
    click on the Move button

```def create_folder(folder_name: str):```
    """Create a new folder in the current application view.
    Args:
        folder_name (str): The name of the folder to create.
    Returns:
        None: Creates a folder with the provided name.
    Examples:
        create_folder("Work Documents")
    """
    click on the add/new button
    click on the Folder icon
    type in the content: folder_name
    click on the Create button | **High-level Instruction**: I believe that the word Yoga Videos is appropriate for use. Rename the yoga folder to Yoga Video.
**- Step 1**: click on the options menu located at the upper right part of the screen.
**- Step 2**: click on the Rename option located at the middle and lower part of the screen.
**- Step 3**: type 'Yoga Video' located at the middle and lower part of the screen.
**- Step 4**: click on the Rename button located at the right and upper part of the screen.
**- Step 5**: stop and set the query as completed.

**High-level Instruction**: I believe that the word Yoga Videos is appropriate for use. Rename the yoga folder to Yoga Video.
**- Step 1**: click on the three-dot menu next to the Yoga folder located at the upper left part of the screen.
**- Step 2**: click on the Rename option located at the right and lower middle part of the screen.
**- Step 3**: click on the delete button located at the bottom right of the screen.
**- Step 4**: type 'Yoga Video' in the input area located at the middle and lower part of the screen.
**- Step 5**: click on the Rename button located at the right and upper middle part of the screen.
**- Step 6**: stop and set the query as completed.

**High-level Instruction**: To keep all yoga videos in one place, I should move a outdoor yoga and home yoga video to the Yoga Video data folder on Google Drive.
**- Step 1**: click on the three-dot menu icon located at the right and bottom part of the screen.
**- Step 2**: scroll up.
**- Step 3**: click on the Move option located at the right and upper middle part of the screen.
**- Step 4**: click on the folder named Yoga Video located at the right and upper part of the screen.
**- Step 5**: click on the Move button located at the bottom right.
**- Step 6**: scroll up.
**- Step 7**: click on the three dots icon located at the right side of the screen.
**- Step 8**: scroll up.
**- Step 9**: click on the Move option located at the right and upper middle part of the screen.
**- Step 10**: click on the Yoga Video folder located at the middle and upper part of the screen.
**- Step 11**: click on the Move button located at the right and lower part of the screen.
**- Step 12**: stop and set the query as completed.

**High-level Instruction**: I think I should create a Yoga folder on Google Drive to keep all of the relevant files in an organized manner.
**- Step 1**: click on the button located at the right and bottom of the screen.
**- Step 2**: click on the Folder icon located at the bottom left.
**- Step 3**: click on the Create button located at the right and lower middle part of the screen.
**- Step 4**: type text 'Yoga' in the activated area.
**- Step 5**: click on the Create button located at the right and lower middle part of the screen.
**- Step 6**: stop and set the query as completed. |

| Rewritten Trajectories |
|---|

**High-level Instruction**: I believe that the word Yoga Videos is appropriate for use. Rename the yoga folder to Yoga Video.
**- Step 1**: click on the options menu located at the upper right part of the screen
**- Step 2**: rename_item("Yoga Video")
**- Step 3**: task complete

**High-level Instruction**: I believe that the word Yoga Videos is appropriate for use. Rename the Yoga folder to Yoga Video.
**- Step 1**: click on the three-dot menu next to the Yoga folder located at the upper left part of the screen
**- Step 2**: rename_item("Yoga Video")
**- Step 3**: task complete

**High-level Instruction**: To keep all yoga videos in one place, I should move an outdoor yoga and home yoga video to the Yoga Video data folder on Google Drive.
**- Step 1**: click on the three-dot menu icon located at the right and bottom part of the screen
**- Step 2**: scroll up
**- Step 3**: move_item_to_folder("Yoga Video")
**- Step 4**: scroll up
**- Step 5**: click on the three dots icon located at the right side of the screen
**- Step 6**: scroll up
**- Step 7**: move_item_to_folder("Yoga Video")
**- Step 8**: task complete

**High-level Instruction**: I think I should create a Yoga folder on Google Drive to keep all of the relevant files in an organized manner.
**- Step 1**: create_folder("Yoga")
**- Step 2**: task complete

Figure 8: Example of meta-function annotation.

| Instruction | Screenshots |
|---|---|

**High-level Instruction**: I'd want to move the Outdoor Yoga and Home Yoga video files to the Yoga Folder in Google Files so they're easier to access in the future.

$Screen_t$      $Screen_{t-1}$

### Action

{"action": "click", "coordinate": [794, 1007]}

### Base Trajectory Augmentation

**Reasoning**: Since the goal is to move the selected Yoga video files to a specific folder, the current screen shows that the 'Move to' option has been selected. The next logical step would be to choose the destination folder where these files should be moved. The highlighted action indicates that the 'Move to' option has been tapped, leading to the next screen where the destination folder can be selected. This aligns with the goal of organizing the files for easier access in the future.

**Low-level Instruction**: click on the 'Move to' option located at the right and upper part of the screen.

**Action**: {"action": "click", "coordinate": [794, 1007]}

### Dynamic Knowledge Construction and Annotation

**Reasoning**: The user has selected two video files, 'Outdoor Yoga.mp4' and 'Home Yoga.mp4', and opened a context menu. The menu offers various options, including 'Move to', which is relevant for organizing files. The user's goal is to move the selected yoga video files to a specific folder. The meta function 'move_item_to_folder' is directly relevant as it deals with moving items to a specified folder. The current screen shows the context menu with a 'Move to' option, which aligns with the meta function's first step. The Similar Trajectories also suggest that selecting 'Move to' is a common step before specifying a folder destination, reinforcing the relevance of this action.

**Low-level Instruction**: click on the 'Move to' option located at the right and upper part of the screen.

**Action**: {"action": "click", "coordinate": [794, 1007]}

Figure 9: Examples of annotations for Base Trajectory Augmentation and Dynamic Knowledge Construction and Annotation. This also serve as training examples. The dynamic knowledge used in the latter is detailed in Fig. 8.

| Instruction | Screenshots |
|---|---|
| **High-level Instruction**: I want to track my garden progress to know how much plants grow in my garden, so track my garden progress in the LeafSnap app. |  |

| Action |
|---|
| {"action": "system_button", "button": "Back"} |

$Screen_t$ $\qquad$ $Screen_{t+1}$

### Base Trajectory Augmentation

**Reasoning**: Since the goal is to track garden progress, pressing back likely takes me to a previous screen where I can find the garden tracking feature. I should navigate back to explore options that allow me to monitor plant growth.
**Low-level Instruction**: press back.
**Action**: {"action": "system_button", "button": "Back"}

### Dynamic Knowledge Construction and Annotation

**Reasoning**: The current screen shows detailed care information for a plant called \'Traveler\'s-palm\', including sections like Water, Fertilizer, Light, Soil, and more. The user appears to be viewing plant details, possibly after identifying a plant or selecting it from a list. The current task is to track garden progress in the LeafSnap app, which aligns with the \'track_garden_progress\' meta function. However, the user is currently viewing detailed information about a specific plant, which suggests they have navigated into a plant detail screen. To track garden progress, they need to return to a broader view of their garden. The \'press back\' instruction likely indicates the need to navigate out of the detailed plant view to access the garden progress or list of plants. The similar trajectory involving \'track_garden_progress\' supports the idea of navigating back to a garden overview.
**Low-level Instruction**: press back.
**Action**: {"action": "system_button", "button": "Back"}

Figure 10: Examples of annotations for Base Trajectory Augmentation and Dynamic Knowledge Construction and Annotation. This also serve as training examples.

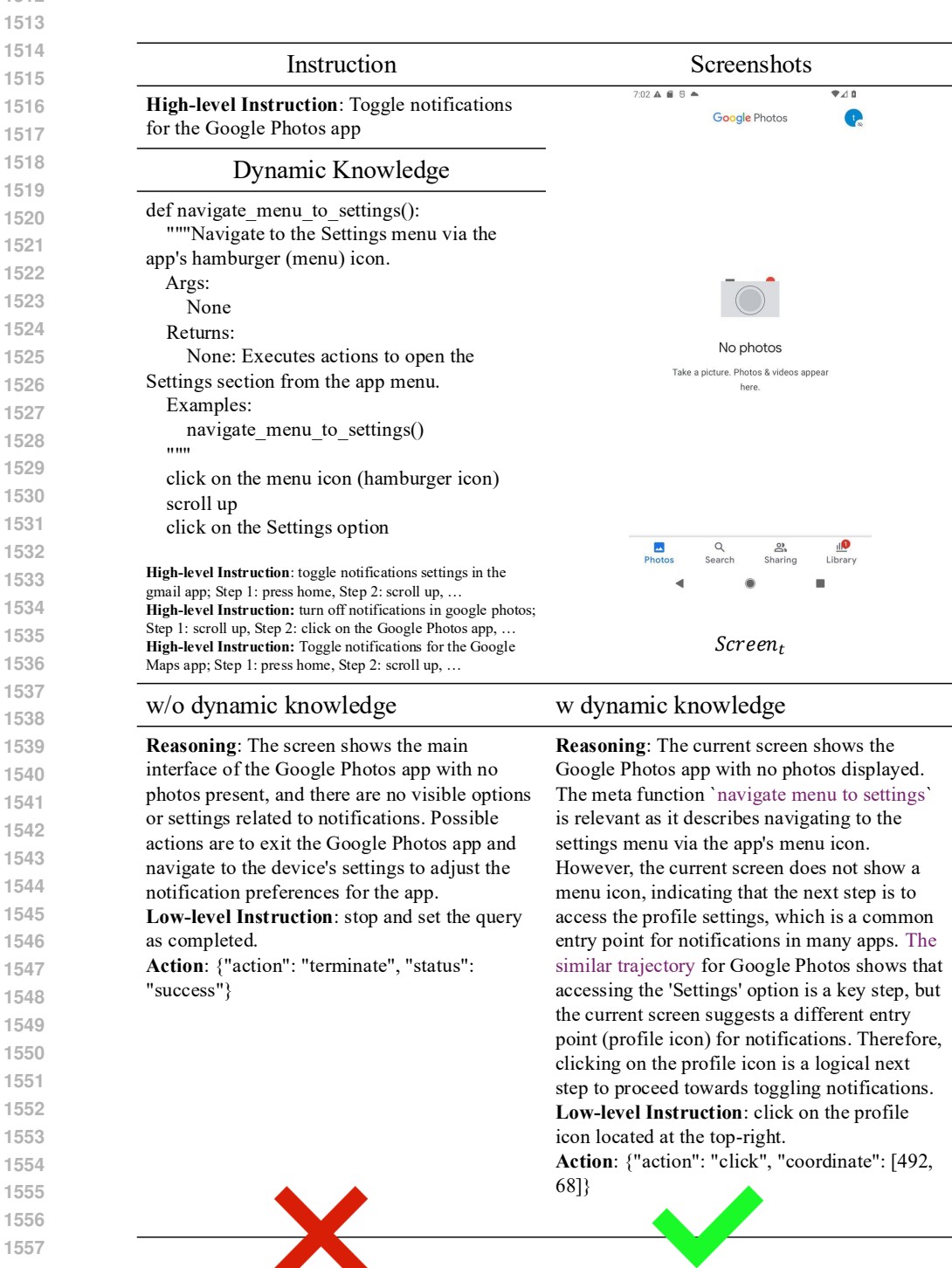

| Instruction | Screenshots |
|---|---|
| **High-level Instruction**: Toggle notifications for the Google Photos app | |

### Dynamic Knowledge

```
def navigate_menu_to_settings():
    """Navigate to the Settings menu via the
app's hamburger (menu) icon.
    Args:
        None
    Returns:
        None: Executes actions to open the
Settings section from the app menu.
    Examples:
        navigate_menu_to_settings()
    """
    click on the menu icon (hamburger icon)
    scroll up
    click on the Settings option
```

**High-level Instruction**: toggle notifications settings in the gmail app; Step 1: press home, Step 2: scroll up, …
**High-level Instruction:** turn off notifications in google photos; Step 1: scroll up, Step 2: click on the Google Photos app, …
**High-level Instruction:** Toggle notifications for the Google Maps app; Step 1: press home, Step 2: scroll up, …

$Screen_t$

| w/o dynamic knowledge | w dynamic knowledge |
|---|---|
| **Reasoning**: The screen shows the main interface of the Google Photos app with no photos present, and there are no visible options or settings related to notifications. Possible actions are to exit the Google Photos app and navigate to the device's settings to adjust the notification preferences for the app.
**Low-level Instruction**: stop and set the query as completed.
**Action**: {"action": "terminate", "status": "success"} | **Reasoning**: The current screen shows the Google Photos app with no photos displayed. The meta function `navigate menu to settings` is relevant as it describes navigating to the settings menu via the app's menu icon. However, the current screen does not show a menu icon, indicating that the next step is to access the profile settings, which is a common entry point for notifications in many apps. The similar trajectory for Google Photos shows that accessing the 'Settings' option is a key step, but the current screen suggests a different entry point (profile icon) for notifications. Therefore, clicking on the profile icon is a logical next step to proceed towards toggling notifications.
**Low-level Instruction**: click on the profile icon located at the top-right.
**Action**: {"action": "click", "coordinate": [492, 68]} |

Figure 11: A case study of DKR-GUI in operation. w/o denotes the variant without dynamic knowledge, while w/ denotes the variant that includes it. The w/ variant often requires a more elaborate thought process to reach the correct decision. Crucially, we observe that the ability to reason with dynamic knowledge is activated during inference, indicating that DKRF endows the model with effective reasoning capabilities in addition to memorizing GUI patterns.

| Instruction | Screenshots |
|---|---|
| **High-level Instruction**: I want to see some Daniel Richter art on the Artsy app for my wife as she is a fan of him. | |

| Dynamic Knowledge | |
|---|---|

def search_in_artsy(search_term: str, category: str = None):
    ......

def share_via_gmail(email_address: str):
    ......

**High-level Instruction**: Open the Artsy app and find an artwork inspired by Andy Warhol; Step 1: Open the artsy app, Step 2: click on search icon, …

**High-level Instruction:** Open the Artsy app and browse paintings.; Step 1: open artsy app, Step 2: Wait, …

| w/o dynamic knowledge | w dynamic knowledge |
|---|---|
| **Reasoning**: The current screen shows a search result for "Daniel Richter" on the Artsy app, with various options like "Top," "Artworks," "Artist," and "Gallery." The goal is to find some of Daniel Richter's art for the user's wife. The next logical step is to select the "Artist" tab to view more detailed information about Daniel Richter, which may include his artworks. This aligns with the goal of exploring his art. **Low-level Instruction**: click on the "Artist" tab located at the upper right part of the screen. **Action**: {"action": "click", "coordinate": [601, 409]} | **Reasoning**: The search term 'Daniel Richter' has been entered, and the search results are currently loading. The screen shows a loading state with placeholders for artwork listings.The current task involves searching for Daniel Richter's art on the Artsy app. The search term has been entered, and the next logical step is to wait for the search results to load. This aligns with the similar trajectory where a search was performed, and the results were awaited. The meta function 'search_in_artsy' is relevant as it involves searching for a term, but the immediate action required is to wait for the page to load, which is a common step after initiating a search. **Low-level Instruction**: wait for the page to load completely. **Action**: {"action": "wait", "time": 3} |

Figure 12: A case study of DKR-GUI in operation. w/o denotes the variant without dynamic knowledge, while w/ denotes the variant that includes it. The w/ variant often requires a more elaborate thought process to reach the correct decision. Crucially, we observe that the ability to reason with dynamic knowledge is activated during inference, indicating that DKRF endows the model with effective reasoning capabilities in addition to memorizing GUI patterns.

| Instruction | Screenshots |
|---|---|
| **High-level Instruction**: In the Smartify Art & Culture app, I'm trying to find a sculpture that was influenced by Solomon Hart so I can find out more about sculpture art. | 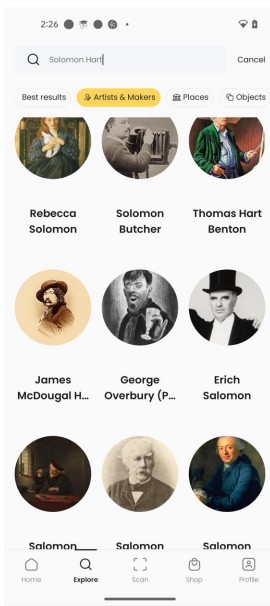 |

| Dynamic Knowledge | |
|---|---|
| def open_app_and_search(app_name: str, search_term: str):
    …… 

def navigate_to_artist(artist_name: str):
    …… 

def scroll_multiple(direction: str, times: int):
    …… 

**High-level Instruction**: In the Smartify Art & Culture app, I want to search for sculpture art since I've been fascinated by it for a long time.; Step 1: Open Smartify app, Step 2: Go to the Explore tab, …
**High-level Instruction:** Pablo Picasso is the artist whose work inspired me, so look for him in the Artier app.; Step 1: Go back to the previous screen, Step 2: Go to the Artists section, … | $Screen_t$ |

| w/o dynamic knowledge | w dynamic knowledge |
|---|---|
| **Reasoning**: The goal is to find a sculpture influenced by Solomon Hart. The current screen shows a list of artists, and the search term 'Solomon Hart' is already entered. The next logical step is to select the correct artist from the search results to explore more about his influence on sculpture.
…
The next action should be to correct the search or select the correct artist to find the desired information.
**Low-level Instruction**: click on the 'Solomon Butcher' option located at the middle and upper part of the screen.
**Action**: {"action": "click", "coordinate": [511.996, 545.952]} | **Reasoning**: The user has searched for 'Solomon Hart' in the Smartify Art & Culture app, and the search results are displayed. The goal is to find a sculpture influenced by Solomon Hart, and the current screen shows various artists, including Solomon Hart, but not necessarily the specific artist the user is looking for. The meta function 'navigate_to_artist' is relevant as it involves selecting an artist from the search results.
…
Therefore, scrolling up is a reasonable action to explore more search results and locate the correct artist.
**Low-level Instruction**: scroll up.
**Action**: {"action": "swipe", "coordinate": [518, 1764], "coordinate2": [518, 804]} |
| 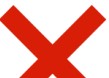 | 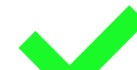 |

Figure 13: A case study of DKR-GUI in operation. w/o denotes the variant without dynamic knowledge, while w/ denotes the variant that includes it. The w/ variant often requires a more elaborate thought process to reach the correct decision. Crucially, we observe that the ability to reason with dynamic knowledge is activated during inference, indicating that DKRF endows the model with effective reasoning capabilities in addition to memorizing GUI patterns.

# F    THE USE OF LARGE LANGUAGE MODELS (LLMS).

We explore the application of MLLMs in GUI agents. Consistent with recent work in the GUI agent field, we integrated MLLMs into the data annotation process to enable automation. Specifically, existing MLLMs were utilized as tools to construct the prompt generation pipeline and to annotate intermediate results. Detailed configurations of the MLLMs employed are provided in the appendix (please refer to section B). Furthermore, during the preparation of this manuscript, we also used MLLMs to assist with language polishing.

