# OpenReview forum: "DKRF: Dynamic Knowledge Reasoning for Out-of-Distribution Generalization in Mobile GUI Agents"
_ICLR.cc/2026/Conference — Submitted to ICLR 2026_

### Official Review · Reviewer_Guzq · 2025-10-17

**Soundness:** 2
**Presentation:** 2
**Contribution:** 2
**Rating:** 4
**Confidence:** 2

**Summary:**

DKRF addresses the poor out-of-distribution generalization of mobile GUI agents, and encourages agents to reason with external “dynamic knowledge” rather than memorizing in-distribution patterns. The authors also develop DKR-GUI, an end-to-end agent fine-tuned using retrieved trajectories and meta-functions, and extend it to MA-DKR where DKR-GUI serves as a planning agent paired with a separate executor. Experiments on AITZ, Android Control, CAGUI, and Kairos indicate improved OOD performance without degrading in-distribution accuracy, and ablations suggest that both knowledge components and decoupling planning from execution contribute to the gains.

**Strengths:**

1. The paper focuses on improving out-of-distribution generalization in mobile GUI agents. Adapting retrieval-augmented reasoning to this domain gives the work relevance to real-world application.
2. The paper presents its problem setup, agent design, and experimental protocol clearly, and it conducts evaluations on both ID and OOD mobile benchmarks. The gains over baseline agents indicate practical value, and the ablations help illustrate the complementary roles of different knowledge sources.

**Weaknesses:**

1. While the paper introduces DKRF and MA-DKR as new frameworks, many of the proposed components appear closely aligned with existing paradigms in RAG-based agents and modular LLM frameworks. Could the authors further clarify which parts of the framework [1][2][3] are genuinely novel in terms of algorithmic design or learning methodology, beyond integrating retrieval into the GUI agent pipeline?
2. Although the paper claims that DKRF enables dynamic knowledge–conditioned reasoning for OOD generalization, the “dynamic knowledge” is still retrieved from pre-existing trajectories within the training corpus. The method relies on static prior data rather than genuinely adapting to unseen task structures or novel interface layouts. It is also unclear whether the model can handle tasks for which no semantically related trajectories exist.

[1] Xu, Ran, et al. "Retrieval-augmented GUI Agents with Generative Guidelines." arXiv preprint arXiv:2509.24183 (2025).
[2] Loo, Gowen, et al. "MobileRAG: Enhancing Mobile Agent with Retrieval-Augmented Generation." arXiv preprint arXiv:2509.03891 (2025).
[3] Li, Yanda, et al. "Appagent v2: Advanced agent for flexible mobile interactions." arXiv preprint arXiv:2408.11824 (2024).

**Questions:**

1. While you mention that the ground-truth thought t^* “must explicitly reference and utilize” the dynamic knowledge D_k, could you clarify how this requirement is actually ensured during data construction or generation? For example, is there any explicit constraint, filtering, or verification mechanism to prevent the model from producing thoughts that overlook or minimally rely on D_k?
2. In cases where the provided dynamic knowledge D_k conflicts with the model’s prior knowledge or pretrained reasoning patterns, how is such inconsistency handled during training or inference? Is there any mechanism to resolve or prioritize between external knowledge and the model’s internal priors?
3. In Table 5, adding a planning agent sometimes brings only marginal or even negative gains, and in a few cases GPT-4o performs worse than the “None” setting. It would be helpful if the authors could clarify what might be driving these outcomes

---

> ### Author Response · Authors · 2025-11-21
> **Response to Reviewer Guzq [1/2]**
>
> **W1**: We appreciate the reviewer’s references. While relevant, RAG-based works (e.g., RAG-GUI, MobileRAG) differ fundamentally from DKRF. Existing methods primarily optimize **external retrieval systems**, premised on the assumption that agents can intrinsically utilize retrieved content. However, our experiments reveal that standard MLLMs struggle with knowledge applicability, alignment, and noise in OOD settings, highlighting a gap between "retrieval" and "utilization." We have incorporated these methodologies into Appendix D (More Related Work) of the revised manuscript.
>
> DKRF addresses this by proposing a **Dynamic Knowledge Reasoning objective**. Rather than merely integrating retrieval, we explicitly train the agent on when to adopt knowledge and how to integrate it with intrinsic experience—an internal reasoning mechanism unexplored in current literature. Validated within the MA-DKR framework, DKRF innovates on **effective knowledge utilization post-retrieval**, offering a distinct and complementary contribution to retrieval-centric approaches.
>
> ---
>
> **W2**: We thank the reviewer for this critical observation. We clarify the relationship between dynamic knowledge and OOD below:
> 1. **Nature of Dynamic Knowledge in OOD**: Although the mechanism is learned, the content remains OOD during evaluation because the interface structures and task definitions (e.g., in CAGUI/Kairos) are unseen. Consequently, the model cannot rely on static priors or template matching; instead, it must employ semantic reasoning to distinguish transferable components within novel contexts. (See "General Reply" for details).
> 2. **Robustness without Relevant Trajectories**: Supplementary experiments confirm that DKRF maintains performance comparable to vanilla SFT even when dynamic knowledge is withheld. This indicates that performance **gains stem from intrinsic knowledge reasoning capabilities** rather than mere reliance on retrieval similarity. Furthermore, DKRF demonstrates significant robustness regarding the relevance of retrieved trajectories in OOD scenarios. We attribute this to DKRF's capability to **extract generalizable features** intrinsic to OOD environments, even from trajectories with **limited relevance**, as shown in Tab. 16.
>
> In summary, utilizing dynamic knowledge mimics real-world behaviors (e.g., consulting tutorials) without compromising OOD authenticity. DKRF trains the model to adapt to "Unknown Task + Unseen UI + Variable External Knowledge," prioritizing robust reasoning over static memorization.
>
> | baseline | FT Method | CAGUI | Kairos | Average |
> | :--- | :--- | :--- | :--- | :--- |
> | Qwen2.5-VL-7B | vallina SFT | 54.4 | 73.2 | 63.8 |
> | | DKRF | 54.0 | 71.5 | 62.8 |
>
> ---
>
> **Q1**: We appreciate the reviewer’s comments on structure. We constrained the output normalization and quality of Qwen2.5-VL-32B during training data construction through multiple measures:
> 1. **Prompt Normalization**: The prompt explicitly mandates the model to output a detailed reasoning process (relevant meta thought) . This instruction dictates the format requirements for knowledge use, requiring the model to explain how the dynamic knowledge guides the instruction, and to rely on its own knowledge when the retrieved context is inapplicable, as shown in Tab. 8.
> 2. **Data Filtering and Validation**: During the extraction pipeline, we enforced strict adherence to JSON formatting and performed automated checks to ensure the existence of the (relevant meta thought) field and its proper correlation with dynamic knowledge (Meta Function/Similar Trajectories).
> 3. **Manual Quality Verification**: We conducted manual verification on 100 randomly selected data samples, confirming that the generated data strictly adhered to the stipulated output format and specifications.
>
> ---
>
> **Q2**: We appreciate the reviewer for highlighting the challenge of conflicting external knowledge and internal priors. DKRF mitigates this via a dual mechanism of **Explicit Conflict Detection** and **Selective Integration**:
> 1. Training Phase: We mandate the teacher model to explicitly evaluate knowledge applicability within the reasoning chain. It must articulate the influence of beneficial knowledge or provide a rationale for rejecting conflicting inputs. This supervision fosters a conflict-aware reasoning paradigm, enabling the model to **dynamically prioritize knowledge** rather than blindly relying on external sources.
> 2. Inference Phase: We employ instruction similarity filtering to mitigate noise. Furthermore, leveraging the "rejection" strategy acquired during training, the model retains the capacity to automatically disregard inapplicable knowledge even if pre-filtering is imperfect.
>
> Thus, rather than functioning as a simplistic switch, DKRF learns an interpretable strategy for knowledge integration, ensuring robust reasoning in OOD scenarios.

---

> ### Author Response · Authors · 2025-11-21
> **Response to Reviewer Guzq [2/2]**
>
> **Q3**: We appreciate the reviewer's attention to the results in Table 5. We clarify herein why certain Planning Agents yield limited gains or, in specific instances, negative effects.
>
> First, Table 5 aims to verify the **compatibility** between different Planning and Executing Agents. In the "Planning Agent = None" setting, Executing Agents (e.g., Qwen2.5-VL-7B, OS-ATLAS-7B) rely on their intrinsic GUI-specialized fine-tuning capabilities for autonomous planning and execution. Since these models already possess robust task understanding and visual grounding capabilities, the baseline performance is inherently strong.
>
> For general-purpose planners like GPT-4o, which are untrained on GUI tasks, their high-level instructions may misalign with executors regarding:
> + Action granularity mismatch (too coarse or too fine);
> + Instruction structural incompatibility with the GUI agent's expected format;
> + Lack of specialized representations for interface elements, thereby introducing noise.
>
> Consequently, on these capable executors, GPT-4o's planning may disrupt their intrinsic optimal policies, leading to marginal negative gains.
>
> In contrast, **DKR-GUI, reinforced by "Dynamic Knowledge Reasoning + GUI-Specialized Training," generates outputs highly consistent with executing agents in structure, granularity, and semantics**. Therefore, DKR-GUI consistently yields improvements regardless of the executor's strength, demonstrating its consistency and high adaptability within the modular framework.

---

> > ### Comment · Reviewer_Guzq · 2025-11-22
> >
> > Thank you for the detailed and thoughtful responses. I appreciate the clarifications and the additional analysis provided by the authors.

---

> > > ### Author Response · Authors · 2025-11-23
> > >
> > > We sincerely thank the reviewer for the constructive feedback. We hope our response has fully addressed your concerns and established a consensus on the contributions of our work. We remain available for any further discussion if you have remaining questions.

---

> > > ### Author Response · Authors · 2025-11-26
> > >
> > > Thank you again for your time and effort in reviewing our work.
> > > We hope that our responses and the new results have addressed your concerns. If so, we would be grateful if you could consider increasing your rating to reflect the improvements made in the revised manuscript.
> > > We remain fully available to answer any other questions you may have.

---

### Official Review · Reviewer_uGHn · 2025-10-24

**Soundness:** 2
**Presentation:** 2
**Contribution:** 3
**Rating:** 4
**Confidence:** 3

**Summary:**

This paper introduces two key contributions: (1) DKRF, a framework for training an MLLM-based agent to call meta functions; and (2) MA-DKR, which leverages the DKRF-trained agent as a planner, augmented with few-shot RAG. The primary novelty lies in enhancing the agent’s meta-level capabilities and knowledge retrieval skills, enabling robust OOD generalization even when trained on a limited dataset.

**Strengths:**

1. The idea is interesting, combing tool-call SFT and few-shot RAG in GUI agent.
2. The paper is clearly written and easy to follow.
3. The ablation experiments are well-organized.

**Weaknesses:**

1. **Insufficient Baseline Comparison**
   The current evaluation lacks several important baselines. It is recommended to include more RL-based models (e.g., UI-R1 [1], GUI-R1 [2], InfiGUI-R1 [3]) as well as representative closed-source models (e.g., GPT and Claude) to better demonstrate DKRF’s capabilities and relative performance.

2. **Benchmark Coverage**
   The evaluation currently omits dynamic mobile benchmarks. In particular, AndroidWorld [4] should be incorporated to more convincingly validate DKRF’s OOD generalization ability.

3. **Related Work Coverage**
   The related work section is incomplete. Additional works on tool-call training (e.g., ToolRL [5], Tool-Star [6], [7], ARPO [8]), GUI tool calls (e.g., CoAct-1 [9]) and GUI few-shot learning (e.g., LearnAct [10]) should be reviewed and discussed. The authors are encouraged to compare their method with these approaches and explicitly highlight DKRF’s novelty in this context.

4. **Experimental Analysis**
   The experimental section could be strengthened by visualizing the call frequency of meta functions before and after DKRF training, to better assess the improvement in meta-function utilization.

5. **Training Details for Reproducibility**
   More details on DKRF’s training process should be provided to enhance reproducibility. Suggested additions include training examples, learning curves.

6. **Paper Presentation**
   The layout of tables (particularly in Appendix C) should be revised. Adjusting table size and placement would improve readability and presentation quality.

---

**References**

[1] UI-R1: Enhancing Efficient Action Prediction of GUI Agents by Reinforcement Learning
[2] GUI-R1: A Generalist R1-Style Vision-Language Action Model For GUI Agents
[3] InfiGUI-R1: Advancing Multimodal GUI Agents from Reactive Actors to Deliberative Reasoners
[4] AndroidWorld: A Dynamic Benchmarking Environment for Autonomous Agents
[5] ToolRL: Reward is All Tool Learning Needs
[6] Tool-Star: Empowering LLM-Brained Multi-Tool Reasoner via Reinforcement Learning
[7] Reinforcing Multi-Turn Reasoning in LLM Agents via Turn-Level Credit Assignment
[8] Agentic Reinforced Policy Optimization
[9] CoAct-1: Computer-using Agents with Coding as Actions
[10] LearnAct: Few-Shot Mobile GUI Agent with a Unified Demonstration Benchmark

**Questions:**

It would be informative to include additional ablations on the training methodology. For example, why not train the meta functions using GRPO directly on the 10K dataset? Such experiments could provide further insights into the design choices and validate the superiority of the proposed approach.

---

If all of the concerns listed above are thoroughly addressed in a revised version, I would be willing to raise my rating score to 6.

---

> ### Author Response · Authors · 2025-11-21
> **Response to Reviewer uGHn [1/2]**
>
> **W1**: We appreciate the reviewer's valuable suggestions. We have systematically supplemented the missing baselines and conducted a unified, reproducible evaluation for both RL-based methods and closed-source models.
>
> First, regarding RL-based baselines: We successfully reproduced GUI-R1 and InfGUI-R1, incorporating them into the main experiments. But we encountered technical dependency issues during the reproduction of UI-R1; thus, this aspect is reserved for future work. Notably, current results indicate that RL methods also generally suffer from limited generalization in OOD GUI tasks, whereas DKRF consistently leads under the same evaluation protocol.
>
> Second, regarding closed-source models (GPT and Claude): Since closed-source models lack a controllable GUI action space and pixel-level instruction alignment capability, we adopted the standard protocol of UGround by integrating them into a unified multi-agent framework for evaluation. We employed Qwen2.5-VL-7B as the Executing Agent. This setup ensures that closed-source models participate fairly in the planning phase while avoiding biases arising from their inherent deficits in visual grounding.
>
> Finally, all baselines (RL-based, planning-based, closed-source/open-source) were evaluated on rigorous OOD datasets (CAGUI, Kairos). The results demonstrate that DKRF achieves the most stable and significant improvements across all model types, further validating the universality and effectiveness of the method itself.
>
> These revisions have been incorporated into the revised manuscript, as shown in Tab. 1-2.
>
> | | AITZ | AC | Average | CAGUI | Kairos | Average |
> | :--- | :--- | :--- | :--- | :--- | :--- | :--- |
> | Claude | 63.3 | 65.3 | 64.3 | 53.8 | 73.1 | 63.5 |
> | GPT | 63.0 | 64.9 | 64.0 | 55.9 | 77.5 | 66.7 |
> | GUI-R1 | 36.7 | 46.6 | 38.6 | 36.9 | 48.8 | 42.9 |
> | InfGUI-R1 | 52.4 | 71.1 | 61.8 | 45.9 | 61.8 | 53.9 |
> | DKR-GUI | 65.1 | 68.6 | 66.9 | 61.2 | 79.6 | **70.4** |
>
> ---
>
> **W2**: We appreciate the reviewer’s valuable suggestion. Since AndroidWorld postdates Qwen2.5-VL, its OOD nature cannot be guaranteed. We recognize the importance of dynamic benchmarks and plan to include this as a supplementary experiment. It was excluded from the current submission due to discrepancies in environment interfaces and action spaces, which necessitate engineering adaptation. We will complete these experiments and update the manuscript (main text and appendix) shortly.
>
> ---
>
> **W3**: We appreciate the reviewer's interest in related works. To avoid conceptual confusion, we further clarify the fundamental differences between DKRF and existing methods. We have incorporated these methodologies into Appendix D (More Related Work) of the revised manuscript.
>
> First, the goal of DKRF is to **enable the GUI Agent to learn how to understand and use external knowledge for dynamic reasoning in real OOD scenarios**. This goal differs significantly from existing methods in task settings, knowledge sources, and reasoning mechanisms, as follows:
> + **LearnAct (Retrieval + ID Environment)**: LearnAct primarily targets In-Distribution (ID) environments, where retrieval conditions are loose and noise is low; the model does not need to perform discriminative reasoning on high-noise trajectories. In contrast, DKRF focuses on OOD few-shot adaptation, performing weakly constrained retrieval based solely on instruction similarity. This compels the model to learn to reason over external knowledge from OOD trajectories with higher noise and more complex information structures, a capability that LearnAct lacks.
> + **CoAct-1 (Cross-Application Automation + Orchestrator)**: CoAct-1 focuses on building system-level collaboration mechanisms for cross-application execution, such as scheduling workflows via code execution and an Orchestrator. Conversely, the core objective of DKRF is to **enhance the model's intrinsic knowledge utilization capability**.
> + **General Tool Use (ARPO, Tool-Star, ToolRL, etc.)**: These works fall under the RL paradigm, improving primarily tool invocation strategies, reward design, or exploration efficiency, with the goal of optimizing "how to invoke tools". DKRF's focus is distinct: we study how the MLLM reasons with external knowledge (such as tool output or retrieved content) after it has been provided, thereby achieving OOD adaptation to GUI environment changes. This is a "knowledge reasoning capability" independent of RL, which current tool-use methods do not address.
>
> In summary, although these methods are formally related to DKRF, fundamental differences exist in research goals, problem settings, and reasoning mechanisms. DKRF aims to solve the enhancement of model reasoning capability under the dual challenge of "Dynamic Knowledge + OOD Environment," a direction not yet covered by existing methods.

---

> ### Author Response · Authors · 2025-11-21
> **Response to Reviewer uGHn [2/2]**
>
> **W4**: We appreciate the reviewer's interest in experimental analysis. We analyze the model's output thought process to verify the explicit invocation of "meta-functions" or "similar trajectories.", as shown in Fig. 5. The results indicate that the frequency with which the model explicitly cites dynamic knowledge significantly increases after DKRF training.
>
> ---
>
> **W5 & W6**: We appreciate the reviewer's reminder and have incorporated the corresponding revisions in the revised manuscript, as shown in Fig. 6 and Fig. 9-13.
>
> ---
>
> **Q1**: We appreciate the reviewer's interest in GRPO. We conducted experiments using GRPO to train GUI agents on DKRF data. Notably, **the lack of supervisory signals for reasoning trajectories impairs the model's ability to utilize dynamic knowledge**. Achieving optimal performance under this paradigm necessitates considerable temporal and computational resources.. Consequently, these findings underscore the practical value of the DKRF methodology, where distillation from stronger MLLMs serves as a more robust and efficient alternative.
>
> | baseline | FT Method | CAGUI | Kairos | Average |
> | :--- | :--- | :--- | :--- | :--- |
> | Qwen2.5-VL-7B | GRPO | 52.3 | 73.7 | 63.0 |
> | | DKRF | 61.2 | 79.6 | 70.4 |

---

> > ### Author Response · Authors · 2025-11-26
> >
> > **W2**: We appreciate the reviewer’s valuable suggestion on dynamic mobile benchmark.
> > We have completed the additional experiments on AndroidWorld as requested.
> > As shown below, DKR-GUI achieves a 27.0% success rate in dynamic mobile environments, significantly outperforming the Qwen2.5-VL-7B baseline (19.0%) with an absolute improvement of 8.0%:
> > | Model | SR(%) |
> > | :--- | :--- |
> > | Qwen2.5-VL-7B | 19.0 |
> > | **DKR-GUI** | **27.0 (+8.0)** |
> >
> > We look forward to hearing your feedback.
> >
> > Thank you.

---

> ### Comment · Reviewer_uGHn · 2025-11-26
>
> Thanks for the author's efforts. I read carefully about your adding experiments, which have resovled most of my concerns. I addmit this work has a good insight to GUI Agent. However, my remaining concerns are still about the novelty (the same as reviewer Guzq). This work seems like an imitation from other domains. Furthermore, although it tells the story more clearly than the original version, the paper writing needs more polishment (e.g., the added figures and case study) for a better presentation and storytelling. So I think this paper is slightly below the ICLR threshold (compared to the GUI Agent paper last year). So I'm going to keep my score.

---

> ### Author Response · Authors · 2025-11-27
>
> Dear Reviewer,
>
> We appreciate your recognition of the "good insight" in our work.
>
> Regarding novelty, we respectfully argue that our approach is fundamentally different from existing studies. Instead of constructing an external **Agentic framework** to enhance performance, we focus on **the core of the Agent—improving the intrinsic capability of the MLLM itself**.
>
> We also accept your feedback on writing and will further polish the paper (including figures and case studies) for better presentation.
>
> Thank you for your valuable opinion.

---

### Official Review · Reviewer_UTCj · 2025-10-29

**Soundness:** 3
**Presentation:** 2
**Contribution:** 3
**Rating:** 4
**Confidence:** 4

**Summary:**

The paper proposes DKRF, a framework that enables GUI agents to generalize to out-of-distribution tasks by retrieving and reasoning over dynamic knowledge from prior trajectories and meta-functions.
Built upon this idea, the authors design DKR-GUI and MA-DKR, achieving strong generalization and state-of-the-art performance across multiple mobile GUI benchmarks.

**Strengths:**

1. The paper studies an interesting and important problem improving the generalization of GUI agents through dynamic knowledge reasoning, which is both novel and practically meaningful.
2. The proposed framework is well-developed and comprehensive and demonstrating solid performance across multiple benchmarks.

**Weaknesses:**

1. The paper suffers from clarity issues in definitions and notation. Many key terms are introduced long before being clearly defined (e.g., in Section 3.2), which makes it difficult to follow the framework. The meanings of several symbols such as $\mathcal{K}$ and $k$ in all $D_k$s​ are unclear, and the paper occasionally mixes $i$ and $I$, which confuses the formulation.

2. The benefit of using meta-function is not well explained. Its role in the overall framework lacks intuitive motivation, and empirically, it seems to contribute only marginal improvement, making it hard to assess its necessity.

3. Although the method is claimed to primarily improve out-of-distribution generalization, the experimental results show similar gains in both in-distribution and OOD settings, which weakens the strength of this claim.

**Questions:**

1. How does the proposed framework handle unseen knowledge in truly OOD tasks? If prior knowledge remains directly applicable to these unseen tasks, does that imply that the OOD setting in the benchmark is not genuinely out-of-distribution?

2. In Table 4, the comparison with other planning-based agents raises fairness concerns. Are all models evaluated under the same conditions (e.g., fine-tuning vs. zero-shot)? Please clarify whether the baselines were evaluated in zero-shot mode, and how this might affect the comparison.

---

> ### Author Response · Authors · 2025-11-21
> **Response to Reviewer UTCj [1/1]**
>
> **W1**:
> We appreciate the reviewer for pointing out the clarity issues. We have made systemic adjustments to the relevant content in the revised manuscript, including:
> + Standardizing the first appearance of key terms to ensure that all concepts are formally and clearly defined before their initial usage;
> + Supplementing and standardizing symbol descriptions, providing a unified explanation for the meanings of $K$, $k$, and all symbols involved in $D_k$;
> + Comprehensively checking and correcting the inconsistent usage of $i$ / $I$ to guarantee the consistency and readability of mathematical expressions.
> These revisions have been incorporated into the revised manuscript, and the coherence of the corresponding paragraphs has been improved.
>
> ---
>
> **W2**: We appreciate the reviewer's valuable feedback regarding the necessity of the meta-function. We wish to clarify our motivation and provide further empirical evidence. These revisions have been incorporated into the revised manuscript in Table 3.
>
> 1. **Motivation: Meta-function aims to "compress long sequences to reduce cost" rather than "pursue maximum performance".**
> The generation of a meta-function refines the original operational trajectory, inevitably leading to some information loss. However, its design objectives are to:
> + Compress long operational sequences into concise, structured functions;
> + Provide task structural information to the model without significantly increasing token cost;
>
> **This approach has proven effective in prior studies [1, 2]** and is widely used for efficient representation of long-sequence tasks.
>
> 2. **Empirical Results: Optimal Trade-off between Performance and Efficiency.**
> We systematically compared the DKR-GUI (end-to-end) and MA-DKR (multi-agent) architectures. As shown in the table below:
> + Meta-functions significantly reduced input token length (only 388 tokens);
> + Performance consistently improved across both architectures compared to the "no dynamic knowledge" setting;
> + Compared to "similar trajectories" (which contain more information but are longer), meta-functions achieved a superior token-performance trade-off;
> + Combining meta-functions with similar trajectories further enhances performance, albeit with increased costs.
>
> Therefore, the meta-function serves not merely as a supplement, but provides essential structural compression capabilities in efficiency-sensitive real-world scenarios.
>
> | Type of Dynamic Knowledge | Average Length of Dynamic Knowledge | Qwen2.5-VL-7B (agent framework) | DKR-GUI (end2end) | MA-DKR (agent framework) |
> | :--- | :--- | :--- | :--- | :--- |
> | None | 0 | 65.4 | 63.8 | 67.1 |
> | Meta-function | 388 | 67.5 | 68.3 | 68.9 |
> | Similar Trajectories | 533 | 67.1 | 70.2 | 69.4 |
> | Similar Trajectories + Meta-function | 921 | 67.9 | 70.4 | 69.9 |
>
> ---
>
> **W3**: We appreciate the reviewer for highlighting this potential misunderstanding. We wish to clarify that **the objective of DKRF is to significantly enhance performance in OOD scenarios without compromising ID performance, rather than simultaneously improving both.**
>
> As shown in Table 3 of the main text:
> + ID Scenarios: DKRF's performance is fundamentally consistent with vanilla SFT, exhibiting only minor fluctuations within the range of normal random variation.
> + OOD Scenarios: DKRF yields substantial and consistent improvements, far exceeding the magnitude of variations observed in ID settings.
> The results in Table 10 of the Appendix further validate this trend.
>
> The fundamental reason is that the model has already encoded knowledge relevant to ID tasks within its parameters. In contrast, the reasoning capability acquired through DKRF is more readily applicable to OOD scenarios; consequently, the primary gains are manifested in the OOD performance.
>
> ---
>
> **Q1**: We appreciate the reviewer's valuable feedback regarding OOD definition. Our research targets OOD scenarios incorporating partial prior knowledge. This setting offers a more realistic simulation of real-world conditions and aligns with established concepts across multiple fields. Please refer to the "General Reply" for a comprehensive explanation.
>
> ---
>
> **Q2**: We apologize for any confusion caused by our presentation. In fact, we conduct comparisons with planning-based agents, as presented in Tables 1, 2, and 3. Crucially, models listed in these tables (excluding the proprietary GPT-4o) have been trained on GUI datasets (e.g., Android Control, AITZ). To ensure a fair evaluation, the training sets for all models strictly excluded the OOD datasets (CAGUI, Kairos) used in our experiments
>
> [1] Ye, Jiabo, et al. "Mobile-agent-v3: Fundamental agents for gui automation." arXiv preprint arXiv:2508.15144 (2025).
>
> [2] Agashe, Saaket, et al. "Agent s: An open agentic framework that uses computers like a human." arXiv preprint arXiv:2410.08164 (2024).

---

> > ### Comment · Reviewer_UTCj · 2025-11-25
> >
> > Thank you for the clarification. The Q3, my question is that since the performance improvements made by DKRF on both in-distribution and out-of-distribution are kind of comparable, the advantage gained by DKRF on ood tasks is mainly coming from the general improvements, which can also be proved by the id situation. Thus, DKRF has no much specefic improvments for the ood tasks. The authors may need to think about the main contribution and story line of the paper.

---

> > > ### Author Response · Authors · 2025-11-26
> > >
> > > We appreciate the reviewer's reply and recognition of the performance gains achieved by DKRF. However, we respectfully wish to **clarify a potential misunderstanding** regarding these improvements. Contrary to the impression of uniform growth, DKRF does not yield comparable performance increases across In-Distribution (ID) and Out-of-Distribution (OOD) scenarios.
> > >
> > > To **disentangle the specific contribution of the DKRF paradigm from the general benefits of instruction tuning**, we conducted a rigorous comparison between DKRF and Vanilla Fine-Tuning (SFT) in our ablation study (**Table 3**). The results demonstrate a clear distinction:
> > > + **On ID Tasks**: DKRF yields **negligible improvement (+0.0% and -0.3%)** over Vanilla FT. This suggests that the pattern memorization capabilities acquired through standard SFT are already sufficient for in-distribution scenarios.
> > > + **On OOD Tasks**: In sharp contrast, DKRF provides a **significant boost (+6.6% and +2.8%)** over Vanilla FT. This empirically demonstrates that our method specifically unlocks the dynamic reasoning capabilities required for unseen, out-of-distribution scenarios.
> > >
> > > Therefore, we emphasize that the primary objective and contribution of DKRF is to enhance robustness and generalization specifically in OOD scenarios, rather than merely providing a general performance uplift.

---

### Official Review · Reviewer_cZN5 · 2025-10-30

**Soundness:** 3
**Presentation:** 3
**Contribution:** 2
**Rating:** 4
**Confidence:** 4

**Summary:**

This paper introduces Dynamic Knowledge Reasoning Fine-tune (DKRF), a paradigm designed to enhance out-of-distribution (OOD) generalization for mobile GUI agents by shifting the learning objective from pattern memorization to knowledge-conditioned reasoning. The authors train an end-to-end agent that explicitly leverages retrieved trajectories and meta-functions during training, and further present a modular variant that combines the agent with a retrieval module and an executing agent. Experiments on four mobile GUI benchmarks show consistent improvements in OOD success rate while maintaining competitive performance on in-distribution tasks.

**Strengths:**

1. The paper targets a well-recognized challenge in GUI agent research, robust OOD generalization, and positions dynamic reasoning as a principled solution beyond scaling or memorization.

2. Results span multiple benchmarks and baselines. Moreover, the proposed method consistently improves OOD performance while preserving ID performance.

**Weaknesses:**

1. The approach relies on a stronger teacher model to generate reasoning traces and dynamic knowledge, which increases computational cost and raises fairness concerns, as external knowledge beyond the original training data is introduced.
2. The reported OS-ATLAS results appear lower than those reported in the original paper (e.g., at least ~71% SR on Android Control in the original results).
3. In real-world scenarios, retrieval may produce irrelevant or low-quality trajectories.. How such knowledge affects the proposed method.
4. The paper lacks qualitative visualization or cases showing how the proposed method plans and executes.
5. The work focuses only on mobile GUI OOD scenarios. It would be useful to clarify whether the method generalizes to desktop GUI environments (e.g., OS-World) and what makes the mobile case special.
6. Can authors report efficiency analyses about the proposed method, e.g., data annotation, training and inference cost, etc.

**Questions:**

Please see the Weakness.

---

> ### Author Response · Authors · 2025-11-21
> **Response to Reviewer cZN5 [1/2]**
>
> **W1**: We fully acknowledge the reviewer's concerns regarding computational cost and fairness. Please refer to the "General Reply" for more details.
>
> **On Computational Cost**
> + One-time Offline Training Cost: The use of a stronger teacher model (Qwen2.5-VL-32B) during data generation represents a one-time, offline computational expense, which is a standard practice in the field .
> + One-time Domain Adaptation Cost: When deploying to a new OOD scenario, the cost associated with a teacher model (e.g., GPT-4o) is incurred only as a one-time expense to construct the knowledge base. Once established, the DKR-GUI operates independently during inference without further reliance on the strong teacher, ensuring efficient deployment .
>
> Our objective is to enhance the model's dynamic reasoning capability, not to gain performance through the accumulation of external knowledge.
>
> **Fairness of Training Data Knowledge Source**: In contrast to vanilla SFT, we did not expand the training set; instead, we replaced 50% of the data.
>
> **Performance improvement stems from the "ability to use knowledge," rather than the "knowledge itself".** We conducted tests by subjecting the baseline model to identical dynamic knowledge conditions. The results indicate:
> + Although the baseline obtained the same knowledge, it failed to utilize it effectively;
> + DKR-GUI performed significantly better under the identical knowledge conditions.
>
> This demonstrates that the performance improvement originates from the knowledge comprehension and reasoning capability learned by the model via DKRF, and not from differences in the knowledge content, as shown in the table below.
>
> | | FT | CAGUI | Kairos | Average |
> | :--- | :--- | :--- | :--- | :--- |
> | Qwen2.5-VL-7B | N/A | 54.7 | 79.0 | 66.9 |
> | Qwen2.5-VL-7B | vanilla SFT | 56.2 | 74.6 | 65.4 |
> | DKR-GUI-7B | DKRF | 61.2 | 79.6 | 70.4 |
>
> ---
>
> **W2**: We appreciate the reviewer for pointing out the inconsistency between the OS-ATLAS results and the original paper. Regarding this point, we wish to provide a clearer explanation: **the performance discrepancy stems entirely from differences in evaluation protocols, rather than an intentional weakening of the baseline. Similar phenomena have also been reported in prior studies [1]**.
> Upon inspecting the OS-ATLAS codebase, we attribute the divergence to two main factors:
> 1. Prompt Discrepancies: The prompts provided by OS-ATLAS are incomplete, particularly regarding the processing of interaction history.
> 2. Evaluation Scope Differences: In the AITZ evaluation, to address the common issue of agents failing to recognize task completion, we explicitly require the model to accurately identify task termination, thereby increasing the challenge.
>
> We reproduced all GUI Agents within this unified evaluation framework to ensure fairness and commit to releasing our code to guarantee reproducibility.
>
> ---
>
> **W3**: We appreciate the reviewer for raising the key question regarding retrieval noise. This issue fundamentally concerns **whether DKRF can maintain stability and reliability when retrieval results are irrelevant or of low quality**. To address this, we introduced a three-layered robustness mechanism in our design:
> 1. **Filtering Mechanism Before Retrieval**. In the DKSeeker, we set a similarity threshold (0.6) to actively filter out obviously irrelevant trajectories.
> 2. **Training for Rejection**: We train the model not only to use knowledge but to critically assess it. As detailed in the prompts in Appendix Table 8, the model is explicitly instructed to reject the retrieved context and rely on its own knowledge if the dynamic knowledge are inapplicable. This supervisory signal equips the model with an intrinsic resistance to noise.
> 3. **Empirical Robustness**: Our experiments (Appendix Table 14) confirm that even when the threshold is lowered to 0.0 (allowing irrelevant inputs), DKR-GUI maintains performance gains without degradation . This demonstrates that the model has learned to extract generalized patterns from OOD contexts rather than being misled by noise.
>
> ---
>
> **W4**:
> We appreciate the reviewer's suggestion regarding visualization. To address this, we have provided comprehensive examples in Appendix **Figures 10-12**. our visualizations depict the model's output mechanism, which consists of three distinct components: Reasoning and Low-level Instruction, which constitute the planning process, and Action, which represents the final execution step. To provide further clarity, we have included additional illustrative examples in the supplementary material.
>
> [1] Zhang, Zhong, et al. "AgentCPM-GUI: Building Mobile-Use Agents with Reinforcement Fine-Tuning." arXiv preprint arXiv:2506.01391 (2025).

---

> ### Author Response · Authors · 2025-11-21
> **Response to Reviewer cZN5 [2/2]**
>
> **W5**: We appreciate the reviewer for raising the important question regarding scenario selection. We think that the mobile environment better exposes the critical challenges required for OOD generalization compared to the desktop or Web environments.
>
> 1. **Mobile Environments Inherently Contain More Real-World OOD Scenarios**:
> Unlike structured and standardized desktop/Web interfaces, the operating environment of mobile devices is highly dynamic and uncontrollable. This is primarily reflected in two aspects:
> + High-Frequency, Unpredictable System Interruptions: Event like Permission pop-ups frequently appear during task execution, instantly creating OOD states unrelated to the task logic, which demands higher stability from the model.
> + Extreme UI Heterogeneity: Desktop/Web typically relies on structural frameworks like DOM/HTML, while the mobile system consists of millions of native Apps, each potentially having independent layout norms, interaction logic, and element styles.
> Based on these characteristics, the mobile setting is more suitable as a rigorous test platform for OOD generalization and better demonstrates the advantages of DKRF
>
> 2. **The Core Design of DKRF Highly Aligns with Mobile OOD Characteristics**:
> DKRF's core objective is to enable the model, when facing highly complex and continuously changing interfaces, to:
> + Be able to timely use dynamic, external, and unseen knowledge;
> + And maintain stable planning capabilities in OOD contexts.
> Therefore, validating DKRF's effectiveness in the mobile environment is the core contribution of this work.
>
> 3. **Regarding Desktop Extension**:
> We agree with the reviewer on the importance of the desktop scene and believe that extending DKRF to desktop GUI is a very meaningful future direction. However, due to significant differences in action space and UI structure between desktop and mobile environments, this process requires non-trivial engineering adaptation work.
>
> ---
>
> **W6**:
> We appreciate the reviewer's question regarding efficiency. This work constructs data and conducts training based on existing mobile GUI datasets in all phases, thus eliminating the need for additional manual annotation overhead. The detailed efficiency analysis is presented as follows:
> 1. **Data-Level Cost**:
> We separately count the data scale, number of screenshots, and input token counts during the annotation process (including without and futher with dynamic knowledge reasoning, as shown in Fig. 4) for three mobile datasets. As shown in the table, the introduction of dynamic knowledge only resulted in a moderate increase in tokens.
>
> 2. **Training Cost**:
> The data amount in the DKRF training phase is identical to that of vanilla SFT. The distinctionis that we replaced 50% of this data with data pairs containing dynamic knowledge, resulting in similar costs.
>
> 3. **Inference Cost**:
> The inference phase of DKR-GUI (7B) does not rely on a teacher model or extra modules. Due to the requirement to process dynamic knowledge, the input tokens slightly increase (20%), and the corresponding FLOPs also show a minor increase (10%), as illustrated in the table.
>
> | Dataset | Tasks Overall | Screenshots | Base Trajectory Augmentation<br>(Input Tokens Pre Step) | Dynamic Knowledge<br>(Tasks w/ dynamic knowledge) | Dynamic Knowledge<br>(Input Tokens Pre Step) |
> | :--- | :--- | :--- | :--- | :--- | :--- |
> | AITZ | 1.0K | 1.0K | 2.3k | 0.5K | 2.3K |
> | Android Control | 6.0K | 35.4K | 7.3k | 3K | 3.9K |
> | GUI Odyssey | 3.0K | 46.2K | 11.0k | 1.5K | 4.3K |
>
> | | FLOPs | Input Tokens Pre Step |
> | :--- | :--- | :--- |
> | vallina SFT | 104.2 | 3.6K |
> | DKRF | 118.8 | 4.4K |

---

> ### Author Response · Authors · 2025-11-26
>
> Dear Reviewer,
>
> I hope this message finds you well. As the discussion period is nearing its end with less than a week remaining, I wanted to ensure we have addressed all your concerns satisfactorily. If there are any additional points or feedback you'd like us to consider, please let us know. Your insights are invaluable to us, and we're eager to address any remaining issues to improve our work.
>
> Thank you for your time and effort in reviewing our paper.

---

### Author Response · Authors · 2025-11-21
**general reply**

We appreciate the reviewer's critical question concerning OOD generalization. We understand the divergence primarily centers on **whether a scenario is still considered "OOD" if a small amount of task-relevant external knowledge is provided during inference.** This issue lacks systematic discussion in GUI Agent, and we hope to clarify our methodology and its necessity. In the revised manuscript, we have incorporated the reviewers' suggestions, and the corresponding modifications are highlighted in blue.

1. **OOD in the GUI Domain is "Dual-Dimensional"; Providing External Knowledge Does Not Invalidate OOD Status**

We categorize OOD generalization into two dimensions:

(1) **Zero-Knowledge OOD**: The traditional definition of OOD:
+ Tasks and UI are out-of-distribution;
+ No supplemental knowledge is provided during inference.

(2) **Knowledge-Adaptive OOD—Our Focus**:

Defined as:
+ Tasks/UIs are out-of-distribution, and the model obtains and dynamically analyzes limited, unseen external knowledge (e.g., similar trajectories, operational examples) during inference to complete the new task.

It is necessary to emphasize that:
+ The tasks and UIs of CAGUI and Kairos were never present in the training set, establishing strict out-of-distribution validity;
+ The "dynamic knowledge" provided during inference is likewise absent in the training data;
Therefore, "OOD under knowledge conditions" is not a relaxation of the OOD criteria but a more accurate characterization of real-world application scenarios.

2. The Open Nature of the GUI Environment Necessitates 'Knowledge-Adaptive OOD'

The GUI environment exhibits the following characteristics:
+ The number of Apps is massive, and versions are updated frequently;
+ The UI state space is complex and continuously drifts;
+ There are a large number of rare elements and long-tail semantics;
+ Complete training set coverage is nearly impossible, and frequent fine-tuning carries high costs and risks (catastrophic forgetting).

As emphasized by UGround [5]:
+ The core difficulty for GUI Agents lies not in the model but in the environment's change rate far exceeding data construction capability.

In most practical applications, humans rely on minimal external references (e.g., tutorials, history records) when confronting new interfaces. Consequently, utilizing small amounts of external knowledge to adapt to OOD environments is more aligned with real-world needs.

**This concept is entirely consistent with the phenomenon of conditional generalization in multimodal models [1], prompt-driven adaptation in ICL [2], and the paradigm of solving OOD problems through external documentation in Retrieval-Augmented Generation (RAG) [3].**

3. DKRF's Core Innovation: Training the Model "How to Use Knowledge," Not Merely "How to Retrieve Knowledge"

Existing RAG/GUI-RAG[6,7] methods typically assume that models can naturally utilize retrieved content during inference, yet extensive research often indicates that models lack stable knowledge integration capability.

To address this, DKRF explicitly constructs during the training phase:

+ Training samples that include dynamic knowledge;
+ Mandating that the reasoning chain forcefully references and integrates this knowledge;
+ Thus enabling the model to directly learn a transferable "knowledge integration function."

This design is consistent with the RAG practice of "teaching the model how to use retrieved content through rationales"[3] , and shares similar logic with "structural OOD generalization that adapts to new target classes through open-set testing" in visual tracking[4].

Therefore, DKRF is not simply adding retrieval; **it explicitly models "how to reason with external knowledge" within the training objective**. This is the fundamental reason the model can improve  in completely unseen UIs/tasks.

4. Modular Framework Further Validates DKRF's Generalizability
We integrated DKR-GUI as the Planning Agent into a UGround-style modular architecture, and experimented with various executing agents.
Results show:
+ Different executing models (including those with weaker grounding capability) benefit significantly;
+ Indicating that the skill learned by DKRF is not a "trick in a certain model," but a reusable, transferable OOD knowledge integration and planning capability.

In sum, this paper's OOD definition guarantees task/UI distribution novelty and explicitly models the universally required "adaptation under external knowledge conditions". DKRF enables stable knowledge-driven reasoning in OOD scenarios by explicitly teaching the model how to understand and integrate dynamic knowledge during training, a point fully supported in ICL, RAG, and visual literature.

---

> ### Author Response · Authors · 2025-11-21
>
> [1] Zhang, Xingxuan, et al. "On the Out-Of-Distribution Generalization of Large Multimodal Models." CVPR2025
>
> [2] Wang Q, Wang Y, Wang Y, et al. Can In-context Learning Really Generalize to Out-of-distribution Tasks? ICLR2025
>
> [3] Wei, Zhepei, Wei-Lin Chen, and Yu Meng. "Instructrag: Instructing retrieval-augmented generation via self-synthesized rationales." ICLR2025
>
> [4] Huang, Lianghua, Xin Zhao, and Kaiqi Huang. "Got-10k: A large high-diversity benchmark for generic object tracking in the wild." TPAMI2019
>
> [5] Gou, Boyu, et al. "Navigating the Digital World as Humans Do: Universal Visual Grounding for GUI Agents." ICLR2025
>
> [6] Xu, Ran, et al. "Retrieval-augmented GUI Agents with Generative Guidelines." arXiv preprint arXiv:2509.24183 (2025).
>
> [7] Loo, Gowen, et al. "MobileRAG: Enhancing Mobile Agent with Retrieval-Augmented Generation." arXiv preprint arXiv:2509.03891 (2025).

---

### Author Response · Authors · 2025-12-02
**Review and Reviewer-Author Discussion Summary**

**Dear PCs, SACs, ACs, and Reviewers,**

We sincerely appreciate your time and valuable contributions to our work. To assist the newly assigned AC and help reduce the workload, we provide below a summary of our contributions, the strengths confirmed by reviewers, and our responses to key concerns during the discussion period.

### **1. Our Contributions**

* **Motivation: Shifting from Memorization to Reasoning.** We identify that existing GUI agents suffer in out-of-distribution (OOD) scenarios due to their reliance on pattern memorization. To address this, we propose the **Dynamic Knowledge Reasoning Fine-tuning (DKRF)** paradigm, which fundamentally shifts the agent's core capability to "dynamic knowledge-based reasoning," allowing it to adapt to unseen tasks by leveraging external knowledge (e.g., similar trajectories).

* **Method: The DKR-GUI Agent & MA-DKR Framework.** We implement this paradigm by training a native agent, **DKR-GUI**, and integrating it into a modular framework, **MA-DKR**. This framework combines DKR-GUI as the core planner (for complex reasoning) with a knowledge retriever and an executing agent (for precise grounding), ensuring both robust planning and accurate execution.

* **Experiments: Superior OOD Generalization & SOTA Performance.** Extensive evaluations demonstrate that MA-DKR achieves **SOTA performance**, improving success rates in challenging OOD scenarios by **9.2%**. Crucially, compared to **vanilla SFT**, our DKRF paradigm achieves a **6.6%** improvement in OOD tasks, empirically validating that dynamic reasoning is a targeted solution that successfully overcomes the "memorization" bottleneck of vanilla fine-tuning.

### **2. Strengths Confirmed by Reviewers**

The reviewers acknowledged the following strengths of our work:

* **Significant & Practical Research Problem:** The paper targets robust OOD generalization, a recognized challenge in GUI agent research. Applying retrieval-augmented reasoning to mobile GUIs is considered highly practical and relevant.
* **Novel Method & Comprehensive Framework:** The proposed **Dynamic Knowledge Reasoning** is recognized as a principled solution beyond simple scaling or memorization. The framework, which cleverly combines tool-call SFT and few-shot RAG, is considered well-developed.
* **Solid Experiments & Significant Gains:** The extensive validation across ID/OOD benchmarks demonstrates solid performance. Key results highlight that the method significantly improves OOD performance while preserving ID capabilities. The well-organized ablation studies clearly illustrate the complementary roles of different knowledge sources.
* **Clear Presentation:** The paper is clearly written, with easy-to-follow definitions, agent design, and experimental protocols.

---

> ### Author Response · Authors · 2025-12-02
>
> ### **3. Responses to Reviewer Concerns**
>
> During the discussion phase, we actively addressed the reviewers' concerns. We believe our responses and revisions have resolved these issues. Specifically:
>
> **Reviewer cZN5**
> * **Experimental Results & Baselines (W2):** We reproduced the mentioned GUI agents under the identical evaluation framework to ensure fair comparison.
> * **Robustness & Generalization (W3, W5):** We demonstrated through experiments that even trajectories with lower task relevance provide gains. We also analyzed the OOD characteristics of mobile scenarios to further validate the advantages of DKRF (more details in **General Reply**).
> * **Cost, Efficiency & Fairness (W1, W6):** We showed that in fair settings with identical dynamic knowledge inputs, DKRF utilizes the knowledge more effectively. We also analyzed cost and efficiency, confirming they remain within acceptable limits.
> * **Visualization (W4):** We added comprehensive case studies, including data annotation and reasoning processes, to the Appendix.
>
> **Reviewer UTCj**
> * **Clarity & Definitions (W1):** We clarified all symbols and definitions in the revised manuscript as requested.
> * **Motivation & Effectiveness (W2):** We experimentally analyzed the trade-off between efficiency and performance provided by meta-functions, demonstrating their effectiveness in specific scenarios.
> * **OOD Definitions & Baselines (W3, Q1):** We detailed our OOD definition with examples from related fields (see **General Reply**). We also clarified a potential misunderstanding: our method improves OOD performance (via DKRF) while maintaining ID performance (via instruction tuning).
> * **Experimental Setup (Q2):** We provided a detailed explanation of our experimental settings.
>
> **Reviewer uGHn**
> * **Evaluation Sufficiency (W1, W2, Q1):** We added the requested baselines (including RL-based models) and compared our method with GRPO to demonstrate the effectiveness of DKRF.
> * **Related Work (W3):** We expanded the comparison with related works (e.g., ToolRL, CoAct-1). We emphasized that **DKRF's novelty lies in enhancing the MLLM's intrinsic reasoning capability**, rather than focusing on external knowledge extraction frameworks (see **General Reply**).
> * **Analysis & Reproducibility (W4, W5):** We added relevant visualizations and details in the revision.
> * **Presentation (W6):** We improved the layout and presentation in the revised version.
>
> **Reviewer Guzq**
> * **Novelty vs. Existing Methods (W1):** We added comparisons and highlighted that DKRF focuses on the **MLLM's internal capability** to utilize knowledge, distinguishing it from works focused solely on retrieval systems (see **General Reply**).
> * **OOD Claims & Benchmark Validity (W2):** We clarified the OOD definition and emphasized that the novelty lies in *how* the knowledge is utilized (reasoning vs. retrieval). We believe this is a novel contribution to the GUI Agent field.
> * **Setup Details & Results (Q1-Q3):** We provided detailed explanations for the setup and results, emphasizing these points in the revised manuscript.
>
> ---
>
> As the above concerns could be effectively addressed through concrete experiments and analyses, we believe our rating would likely have been raised to `6664` had this unexpected situation not occurred. We sincerely thank the reviewers, AC, SAC, and PC for their dedicated efforts during this challenging period, and we hope that this summary is helpful for the upcoming decision phase.
>
>
>
> **Best wishes to all!**

---

### Meta-Review · Area_Chair_bNp9 · 2026-01-06

**Summary:**

The paper presents a framework for grounded LLM-based mobile GUI agents that combines (1) retrieving trajectories from the training set (2) "meta-functions" which summarize sub-tasks from multiple similar trajectories and (3) a modular framework with a retrieval module and an executing agent. The paper evaluates on multiple mobile benchmarks, improving performance in OOD generalization in comparison to baselines.

Strengths: Reviewers appreciated the significance and practicality of the tasks and the focus on OOD generalization. They also felt that retrieval, and training the model to use the retrieved knowledge, were well-motivated. Most reviewers acknowledged the solid performance improvements. While there were concerns about comparison methods, the author response adequately (in my mind) addressed these.

**Reviewer Concerns:**

A main remaining concern after the discussion is the novelty in comparison to other works on GUI agents.  While other papers also introduce these components, the main novelties here seems to be the combination of the components, training the agent model to use the retrieved knowledge (what the author response describes as "enhancing the MLLM's intrinsic reasoning capability"), and the focus on OOD generalization.

**Reviewer Scores:**

Three reviewers acknowledged the author response without raising their score; the other [cZN5] did not reply. I predict that the final scores would have been 6/4/4/4 (one reviewer would have raised their score, probably cZN5).

---

### Decision · Program_Chairs · 2026-01-26

Reject